# MANIFOLD DIFFUSION FIELDS

**Ahmed A. Elhag** [*]**, Yuyang Wang, Joshua M. Susskind, Miguel Angel Bautista**
Apple
`{aa_elhag, yuyang_wang4, jsusskind, mbautistamartin}@apple.com`

## ABSTRACT

We present Manifold Diffusion Fields (MDF), an approach that unlocks learning of diffusion models of data in general non-Euclidean geometries. Leveraging insights from spectral geometry analysis, we define an intrinsic coordinate system on the manifold via the eigen-functions of the Laplace-Beltrami Operator. MDF represents functions using an explicit parametrization formed by a set of multiple input-output pairs. Our approach allows to sample continuous functions on manifolds and is invariant with respect to rigid and isometric transformations of the manifold. In addition, we show that MDF generalizes to the case where the training set contains functions on different manifolds. Empirical results on multiple datasets and manifolds including challenging scientific problems like weather prediction or molecular conformation show that MDF can capture distributions of such functions with better diversity and fidelity than previous approaches.

## 1 INTRODUCTION

Approximating probability distributions from finite observational datasets is a pivotal machine learning challenge, with recent strides made in areas like text (Brown et al., 2020), images (Nichol & Dhariwal, 2021), and video (Ho et al., 2022). The burgeoning interest in diffusion generative models (Ho et al., 2020; Nichol & Dhariwal, 2021; Song et al., 2021b) can be attributed to their stable optimization goals and fewer training anomalies (Kodali et al., 2017). However, fully utilizing the potential of these models across scientific and engineering disciplines remains an open problem. While diffusion generative models excel in domains with Euclidean (*i.e.* flat) spaces like 2D images or 3D geometry and video, many scientific problems involve reasoning about continuous functions on curved spaces (*i.e.* Riemannian manifolds). Examples include climate observations on the sphere (Hersbach et al., 2020; Lindgren et al., 2011) or solving PDEs on curved surfaces, which is a crucial problem in areas like quantum mechanics (Bhabha, 1945) and molecular conformation (Jing et al., 2022). Recent works have tackled the problem of learning generative models of continuous functions following either adversarial formulations (Dupont et al., 2022b), latent parametrizations (Dupont et al., 2022a; Du et al., 2021; Bauer et al., 2023), or diffusion models (Bond-Taylor & Willcocks, 2023; Zhuang et al., 2023). While these approaches have shown promise on functions within the Euclidean domain, the general case of learning generative models of functions on Riemannian manifolds remains unexplored.

In this paper, we introduce Manifold Diffusion Fields (MDF), extending generative models over functions to the Riemannian setting. We take the term *function* and *field* to have equivalent meaning throughout the paper. Note that these are not to be confused with gradient vector fields typically used on manifold. These fields $f : \mathcal{M} \to \mathcal{Y}$ map points from a manifold $\mathcal{M}$ (that might be parametrized as a 3D mesh, graph or even a pointcloud, see Sect. 5.2) to corresponding values in signal space $\mathcal{Y}$. MDF is trained on collections of fields and learns a generative model that can sample different fields over a manifold. In Fig. 1 we show real samples of such functions for different manifolds, as well as samples generated by MDF.

Here are our main contributions:

- We borrow insights from spectral geometry analysis to define a coordinate system for points in manifolds using the eigen-functions of the Laplace-Beltrami Operator.

---

[*]Work was completed while A.A.E was an intern with Apple.

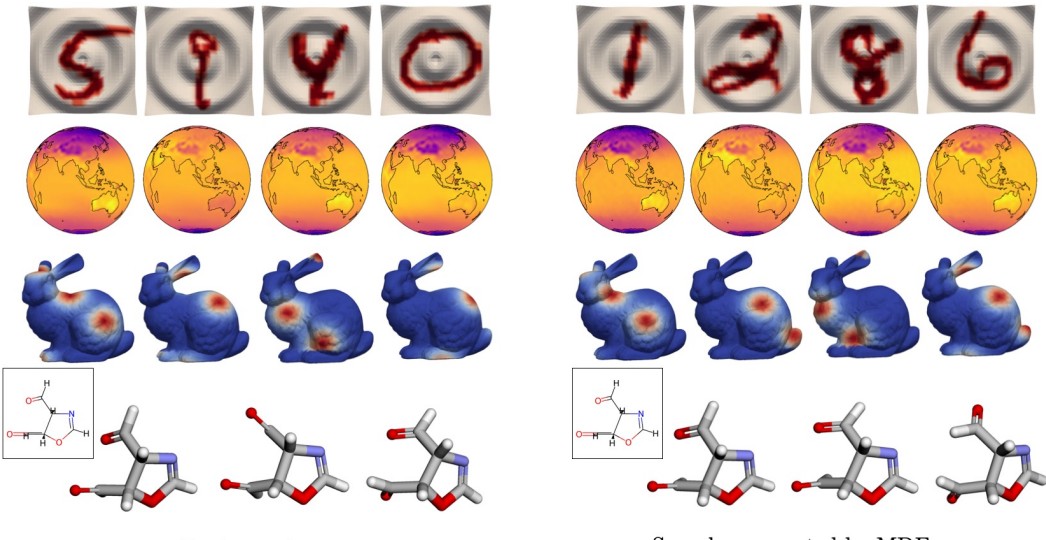

Real samples                    Samples generated by MDF

Figure 1: MDF learns a distribution over a collection of fields $f : \mathcal{M} \rightarrow \mathbb{R}^d$, where each field is defined on a manifold $\mathcal{M}$. We show real samples and MDF's generations on different datasets of fields defined on different manifolds. **First row:** MNIST digits on the sine wave manifold. **Second row Middle:** ERA5 climate dataset (Hersbach et al., 2020) on the 2D sphere. **Third row:** GMM dataset on the bunny manifold. **Fourth row:** molecular conformations in GEOM-QM9 (Ruddigkeit et al., 2012) given the molecular graph.

- We formulate an end-to-end generative model for functions defined on manifolds, allowing sampling different fields over a manifold. Focusing on practical settings, our extensive experimental evaluation covers graphs, meshes and pointclouds as approximations of manifolds.

- Our model outperforms recent approaches like (Zhuang et al., 2023; Dupont et al., 2022b), yielding diverse and high fidelity samples, while being robust to rigid and isometric manifold transformations. Results on climate modeling datasets (Hersbach et al., 2020) and PDE problems show the practicality of MDF in scientific domains.

- On the challenging problem of molecular conformer generation, MDF obtains state-of-the-art results on GEOM-QM9 (Ruddigkeit et al., 2012).

## 2 RELATED WORK

Our approach extends recent efforts in generative models for continuous functions in Euclidean space (Zhuang et al., 2023; Dupont et al., 2022b;a; Du et al., 2021), shown Fig. 2(a), to functions defined over manifolds, see Fig. 2(b). The term Implicit Neural Representation (INR) is used in these works to denote a parameterization of a single function (*e.g.* a single image in 2D) using a neural network that maps the function's inputs (*i.e.* pixel coordinates) to its outputs (*i.e.* RGB values). Different approaches have been proposed to learn distributions over fields in Euclidean space, GASP (Dupont et al., 2022b) leverages a GAN whose generator produces field data whereas a point cloud discriminator operates on discretized data and aims to differentiate real and generated functions. Two-stage approaches (Dupont et al., 2022a; Du et al., 2021) adopt a latent field parameterization (Park et al., 2019) where functions are parameterized via a hyper-network (Ha et al., 2017) and a generative model is learnt on the latent or INR representations. In addition, MDF also relates to recent work focusing on fitting a function (*e.g.* learning an INR) on a manifold using an intrinsic coordinate system (Koestler et al., 2022; Grattarola & Vandergheynst, 2022), and generalizes it to the problem of learning a probabilistic model over multiple functions defined on a manifold.

Intrinsic coordinate systems have also been recently used in the context of Graph Transformers(Maskey et al., 2022; Sharp et al., 2022; He et al., 2022; Dwivedi et al., 2020), where eigenvectors of the Graph Laplacian are used to replace standard positional embeddings (in addition to also

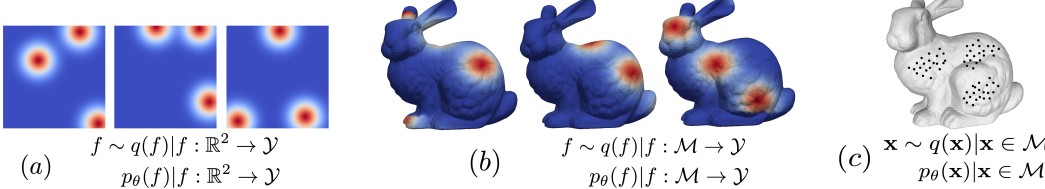

Figure 2: (a) Generative models of fields in Euclidean space (Zhuang et al., 2023; Dupont et al., 2022b;a; Du et al., 2021) learn a distribution $p_\theta$ over **functions whose domain is** $\mathbb{R}^n$. We show an example where each function is the result of evaluating a Gaussian mixture with 3 random components in 2D. (b) MDF learns a distribution $p_\theta$ from a collection of **fields whose domain is a general Riemannian manifold** $f \sim q(f)|f : \mathcal{M} \to \mathcal{Y}$. Similarly, as an illustrative example each function is the result of evaluating a Gaussian mixture with 3 random components on $\mathcal{M}$ (*i.e.* the Stanford bunny). (c) Riemannian generative models (Bortoli et al., 2022; Gemici et al., 2016; Rozen et al., 2021; Chen & Lipman, 2023) learn a parametric distribution $p_\theta$ from an empirical observations $x \sim q(x)|x \in \mathcal{M}$ of **points $x$ on a Riemannian manifold** $\mathcal{M}$, denoted by black dots on the manifold.

using edge features). In this setting, Graph Transformer architectures have been used for supervised learning problems like graph/node classification and regression, whereas we focus on generative modeling.

The learning problem we tackle with MDF can be interpreted as lifting the Riemannian generative modeling problem (Bortoli et al., 2022; Gemici et al., 2016; Rozen et al., 2021; Chen & Lipman, 2023) to function spaces. Fig. 2(b)(c) show the training setting for the two problems, which are related but not directly comparable. MDF learns a generative model over functions defined on manifolds, *e.g.* a probability density over functions $f : \mathcal{M} \to \mathcal{Y}$ that map points in the manifold $\mathcal{M}$ to a signal space $\mathcal{Y}$. In contrast, the goal in Riemannian generative modeling is to learn a probability density from an observed set of points living in a Riemannian manifold $\mathcal{M}$. For example, in the case of the bunny, shown in Fig. 2(c), a Riemannian generative model learns a distribution of points $x \in \mathcal{M}$ on the manifold.

MDF is also related to work on Neural Processes (Garnelo et al., 2018; Kim et al., 2019; Dutordoir et al., 2022), which also learn distributions over functions. As opposed to the formulation of Neural Processes which optimizes an ELBO (Kingma & Welling, 2014) we formulate MDF as a denoising diffusion process in function space, which results in a robust training objective and a powerful inference process. Moreover, our work relates to formulations of Gaussian Processes (GP) on Riemannian manifolds (Borovitskiy et al., 2020; Hutchinson et al., 2021). These approaches are GP formulations of Riemannian generative modeling (see Fig. 2), in the sense that they learn conditional distributions of points on the manifold, as opposed to distributions over functions on the manifold like MDF.

## 3 PRELIMINARIES

### 3.1 DENOISING DIFFUSION PROBABILISTIC MODELS

Denoising Diffusion Probabilistic Models (Ho et al., 2020) (DDPMs) belong to the broad family of latent variable models. We refer the reader to (Everett, 2013) for an in depth review. In short, to learn a parametric data distribution $p_\theta(x_0)$ from an empirical distribution of finite samples $q(x_0)$, DDPMs reverse a diffusion Markov Chain that generates latents $x_{1:T}$ by gradually adding Gaussian noise to the data $x_0 \sim q(x_0)$ for $T$ time-steps as follows: $q(x_t|x_{t-1}) := \mathcal{N}(x_{t-1}; \sqrt{\bar{\alpha}_t}x_0, (1 - \bar{\alpha}_t)\mathbf{I})$. Here, $\bar{\alpha}_t$ is the cumulative product of fixed variances with a handcrafted scheduling up to time-step $t$. (Ho et al., 2020) introduce an efficient training recipe in which: i) The forward process adopts sampling in closed form. ii) reversing the diffusion process is equivalent to learning a sequence of denoising (or score) networks $\epsilon_\theta$, with tied weights. Reparameterizing the forward process as $x_t = \sqrt{\bar{\alpha}_t}x_0 + \sqrt{1 - \bar{\alpha}_t}\epsilon$ results in the "simple" DDPM loss: $\mathbb{E}_{t\sim[0,T],x_0\sim q(x_0),\epsilon\sim\mathcal{N}(0,\mathbf{I})}\left[\|\epsilon - \epsilon_\theta(\sqrt{\bar{\alpha}_t}x_0 + \sqrt{1 - \bar{\alpha}_t}\epsilon, t)\|^2\right]$, which makes learning of the data distribution $p_\theta(x_0)$ both efficient and scalable.

At inference time, we compute $\boldsymbol{x}_0 \sim p_\theta(\boldsymbol{x}_0)$ via ancestral sampling (Ho et al., 2020). Concretely, we start by sampling $\boldsymbol{x}_T \sim \mathcal{N}(\boldsymbol{0}, \mathbf{I})$ and iteratively apply the score network $\epsilon_\theta$ to denoise $\boldsymbol{x}_T$, thus reversing the diffusion Markov Chain to obtain $\boldsymbol{x}_0$. Sampling $\boldsymbol{x}_{t-1} \sim p_\theta(\boldsymbol{x}_{t-1}|\boldsymbol{x}_t)$ is equivalent to computing the update: $\boldsymbol{x}_{t-1} = \frac{1}{\sqrt{\alpha_t}}\left(\boldsymbol{x}_t - \frac{1-\alpha_t}{\sqrt{1-\alpha_t}}\epsilon_\theta(\boldsymbol{x}_t, t)\right) + \mathbf{z}$, where at each inference step a stochastic component $\mathbf{z} \sim \mathcal{N}(\boldsymbol{0}, \mathbf{I})$ is injected, resembling sampling via Langevin dynamics (Welling & Teh, 2011). In practice, DDPMs have obtained amazing results for signals living in an Euclidean grid (Nichol & Dhariwal, 2021; Ho et al., 2022). However, the extension to functions defined on curved manifolds remains an open problem.

## 3.2 RIEMANNIAN MANIFOLDS

Previous work on Riemannian generative models (Bortoli et al., 2022; Gemici et al., 2016; Rozen et al., 2021; Chen & Lipman, 2023) develops machinery to learn distribution from a training set of points living on Riemannian manifolds. In this work, we assume manifolds are compact and connected Riemannian manifolds $\mathcal{M}$ equipped with a smooth metric $g : T_{\boldsymbol{x}}\mathcal{M} \times T_{\boldsymbol{x}}\mathcal{M} \to \mathbb{R}_{\geq 0}$ (*e.g.* a smoothly varying inner product from which a distance can be constructed on $\mathcal{M}$). A core tool in Riemannian manifolds is the tangent space, this space defines the tangent hyper-plane of a point $\boldsymbol{x} \in \mathcal{M}$ and is denoted by $T_{\boldsymbol{x}}\mathcal{M}$. This tangent space $T_{\boldsymbol{x}}\mathcal{M}$ is used to define inner products $\langle \boldsymbol{u}, \boldsymbol{v} \rangle_g, \boldsymbol{u}, \boldsymbol{v} \in T_{\boldsymbol{x}}\mathcal{M}$, which in turns defines $g$. The tangent bundle $T\mathcal{M}$ is defined as the collection of tangent spaces for all points $T_{\boldsymbol{x}}\mathcal{M} \; \forall \boldsymbol{x} \in \mathcal{M}$.

In practice we cannot assume that for general geometries (*e.g.* geometries for which we don't have access to a closed form and are commonly represented as graphs/meshes) one can efficiently compute $g$. While it is possible to define an analytical form for the Riemannian metric $g$ on simple parametric manifolds (*e.g.* hyper-spheres, hyperbolic spaces, tori), general geometries (*i.e.* the Stanford bunny) are inherently discrete and irregular, which can make it expensive to even approximate $g$. To mitigate these issues MDF is formulated from the ground up without relying on access to an analytical form for $g$ or the tangent bundle $T\mathcal{M}$ and allows for learning a distribution of functions defined on general geometries.

## 3.3 LAPLACE-BELTRAMI OPERATOR

The Laplace-Beltrami Operator (LBO) denoted by $\Delta_{\mathcal{M}}$ is one of the cornerstones of differential geometry and can be intuitively understood as a generalization of the Laplace operator to functions defined on Riemannian manifolds $\mathcal{M}$. Intuitively, the LBO encodes information about the curvature of the manifold and how it bends and twists at every point, reflecting the intrinsic geometry. One of the basic uses of the Laplace-Beltrami operator is to define a functional basis on the manifold by solving the general eigenvalue problem associated with $\Delta_{\mathcal{M}}$, which is a foundational technique in spectral geometry analysis (Lévy, 2006). The eigen-decomposition of $\Delta_{\mathcal{M}}$ are the non-trivial solutions to the equation $\Delta_{\mathcal{M}}\varphi_i = \lambda_i\varphi_i$. The eigen-functions $\varphi_i : \mathcal{M} \to \mathbb{R}$ represent an orthonormal functional basis for the space of square integrable functions (Lévy, 2006; Minakshisundaram & Pleijel, 1949). Thus, one can express a square integrable function $f : \mathcal{M} \to \mathcal{Y}$, with $f \in L^2$ as a linear combination of the functional basis, as follows: $f = \sum\limits_{i=1}^{\infty} \langle f, \varphi_i \rangle \varphi_i$.

In practice, the infinite sum is truncated to the $k$ eigen-functions with lowest eigen-values, where the ordering of the eigen-values $\lambda_1 < \lambda_2 \cdots < \lambda_k$ enables a low-pass filter of the basis. Moreover, (Lévy, 2006) shows that the eigen-functions of $\Delta_M$ can be interpreted as a Fourier-like function basis (Vallet & Lévy, 2008) on the manifold, *e.g.* an intrinsic coordinate system for the manifold. In particular, if $\mathcal{M} = S^2$ this functional basis is equivalent to spherical harmonics, and in Euclidean space it becomes a Fourier basis which is typically used in implicit representations (Xie et al., 2022). MDF uses the eigen-functions of the LBO $\Delta_{\mathcal{M}}$ to define a Fourier-like positional embedding (PE) for points on $\mathcal{M}$ (see Fig. 3). Note that these eigen-functions are only defined for points that lie on the manifold, making MDF strictly operate on the manifold.

## 4 METHOD

MDF is a diffusion generative model that captures distributions over fields defined on a Riemannian manifold $\mathcal{M}$. We are given observations in the form of an empirical distribution $f_0 \sim q(f_0)$ over

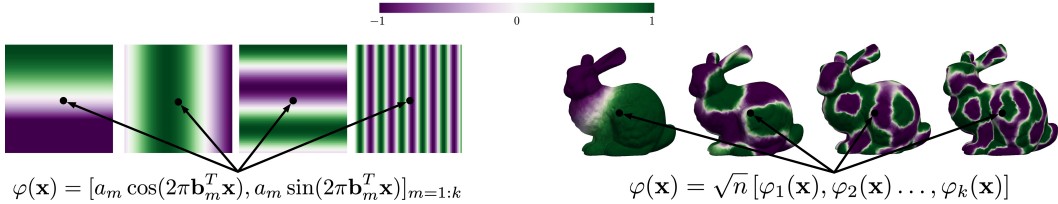

$$\varphi(\mathbf{x}) = [a_m \cos(2\pi \mathbf{b}_m^T \mathbf{x}), a_m \sin(2\pi \mathbf{b}_m^T \mathbf{x})]_{m=1:k} \qquad \varphi(\mathbf{x}) = \sqrt{n}\,[\varphi_1(\mathbf{x}), \varphi_2(\mathbf{x}) \ldots, \varphi_k(\mathbf{x})]$$

Figure 3: **Left:** Fourier PE of a point $\boldsymbol{x}$ in 2D Euclidean space. Generative models of functions in ambient space (Zhuang et al., 2023; Dupont et al., 2022b;a; Du et al., 2021) use this representation to encode a function's input. **Right:** MDF uses the eigen-functions $\varphi_i$ of the Laplace-Beltrami Operator (LBO) $\Delta_{\mathcal{M}}$ evaluated at a point $\boldsymbol{x} \in \mathcal{M}$.

fields where a field $f_0 : \mathcal{M} \to \mathcal{Y}$ maps points from a manifold $\mathcal{M}$ to a signal space $\mathcal{Y}$. As a result, latent variables $f_{1:T}$ are also fields on manifolds that can be continuously evaluated.

To tackle the problem of learning a diffusion generative model over fields we employ a similar recipe to (Zhuang et al., 2023), generalizing from fields defined on Euclidean domains to functions on Riemannian manifolds. In order to this we use the first $k$ eigen-functions $\varphi_{i=1:k}$ of $\Delta_M$ to define a Fourier-like representation on $\mathcal{M}$. Note that our model is *independent of the particular parametrization of the LBO*, *e.g.* cotangent, point cloud (Sharp & Crane, 2020) or graph laplacians can be used depending on the available manifold parametrization (see Sect. 5.2 for experimental results). We use the term $\varphi(\boldsymbol{x}) = \sqrt{n}[\varphi_1(\boldsymbol{x}), \varphi_2(\boldsymbol{x}), \ldots, \varphi_k(\boldsymbol{x})] \in \mathbb{R}^k$ to denote the normalized eigen-function representation of a point $\boldsymbol{x} \in \mathcal{M}$. In Fig. 3 we show a visual comparison of standard Fourier PE on Euclidean space and the eigen-functions of the LBO on a manifold.

We adopt an explicit field parametrization (Zhuang et al., 2023), where a field is characterized by a set of coordinate-signal pairs $\{(\varphi(\boldsymbol{x}_c), \boldsymbol{y}_{(c,0)})\}$, $\boldsymbol{x}_c \in \mathcal{M}, \boldsymbol{y}_{(c,0)} \in \mathcal{Y}$, which is denoted as *context set*. We row-wise stack the context set and refer to the resulting matrix via $\mathbf{C}_0 = [\varphi(\mathbf{X}_c), \mathbf{Y}_{(c,0)}]$. Here, $\varphi(\mathbf{X}_c)$ denotes the eigen-function representation of the coordinate portion and $\mathbf{Y}_{(c,0)}$ denotes the signal portion of the context set at time $t = 0$. We define the forward process for the context set by diffusing the signal and keeping the eigen-functions fixed:

$$\mathbf{C}_t = [\varphi(\mathbf{X}_c), \mathbf{Y}_{(c,t)} = \sqrt{\bar{\alpha}_t}\mathbf{Y}_{(c,0)} + \sqrt{1 - \bar{\alpha}_t}\epsilon_c], \tag{1}$$

where $\epsilon_c \sim \mathcal{N}(\mathbf{0}, \mathbf{I})$ is a noise vector of the appropriate size. We now turn to the task of formulating a score network for fields. Following (Zhuang et al., 2023), the score network needs to take as input the context set (*i.e.* the field parametrization), and needs to accept being evaluated continuously in $\mathcal{M}$. We do this by employing a *query set* $\{\boldsymbol{x}_q, \boldsymbol{y}_{(q,0)}\}$. Equivalently to the context set, we row-wise stack query pairs and denote the resulting matrix as $\mathbf{Q}_0 = [\varphi(\mathbf{X}_q), \mathbf{Y}_{(q,0)}]$. Note that the forward diffusion process is equivalently defined for both context and query sets:

$$\mathbf{Q}_t = [\varphi(\mathbf{X}_q), \mathbf{Y}_{(q,t)} = \sqrt{\bar{\alpha}_t}\mathbf{Y}_{(q,0)} + \sqrt{1 - \bar{\alpha}_t}\epsilon_q], \tag{2}$$

where $\epsilon_q \sim \mathcal{N}(\mathbf{0}, \mathbf{I})$ is a noise vector of the appropriate size. The underlying field is solely defined by the context set, and the query set are the function evaluations to be de-noised. The resulting *score field* model is formulated as follows, $\hat{\epsilon}_q = \epsilon_\theta(\mathbf{C}_t, t, \mathbf{Q}_t)$.

Using the explicit field characterization and the score field network, we obtain the training and inference procedures in Alg. 1 and Alg. 2, respectively, which are accompanied by illustrative examples of sampling a field encoding a Gaussian mixture model over the manifold (*i.e.* the bunny). For training, we uniformly sample context and query sets from $f_0 \sim \text{Uniform}(q(f_0))$ and only corrupt their signal using the forward process in Eq. equation 1 and Eq. equation 2. We train the score field network $\epsilon_\theta$ to denoise the signal portion of the query set, given the context set. During sampling, to generate a field $f_0 \sim p_\theta(f_0)$ we first define a query set $\mathbf{Q}_T = [\varphi(\mathbf{X}_q), \mathbf{Y}_{(q,T)} \sim \mathcal{N}(\mathbf{0}, \mathbf{I})]$ of random values to be de-noised. Similar to (Zhuang et al., 2023) we set the context set to be a random subset of the query set. We use the context set to denoise the query set and follow ancestral sampling as in the vanilla DDPM (Ho et al., 2020). Note that during inference the eigen-function representation $\varphi(x)$ of the context and query sets does not change, only their corresponding signal value.

**Algorithm 1** Training

1: $\Delta_{\mathcal{M}}\varphi_i = \varphi_i \lambda_i$ // LBO eigen-decomposition
2: **repeat**
3:   $(\mathbf{C}_0, \mathbf{Q}_0) \sim \text{Uniform}(q(f_0))$
4:   $t \sim \text{Uniform}(\{1, \ldots, T\})$
5:   $\epsilon_c \sim \mathcal{N}(\mathbf{0}, \mathbf{I}), \epsilon_q \sim \mathcal{N}(\mathbf{0}, \mathbf{I})$
6:   $\mathbf{C}_t = [\varphi(\mathbf{X}_c), \sqrt{\bar{\alpha}_t}\mathbf{Y}_{(c,0)} + \sqrt{1-\bar{\alpha}_t}\epsilon_c]$
7:   $\mathbf{Q}_t = [\varphi(\mathbf{X}_q), \sqrt{\bar{\alpha}_t}\mathbf{Y}_{(q,0)} + \sqrt{1-\bar{\alpha}_t}\epsilon_q]$
8:   Take gradient descent step on
    $\nabla_\theta \|\epsilon_q - \epsilon_\theta(\mathbf{C}_t, t, \mathbf{Q}_t)\|^2$
9: **until** converged

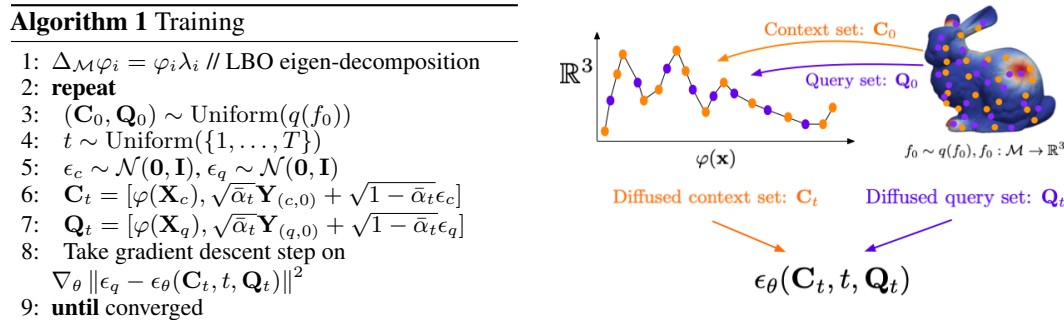

Figure 4: **Left:** MDF training algorithm. **Right**: Visual depiction of a training iteration for a field on the bunny manifold $\mathcal{M}$. See Sect. 4 for definitions.

**Algorithm 2** Sampling

1: $\Delta_{\mathcal{M}}\varphi_i = \varphi_i \lambda_i$ // LBO eigen-decomposition
2: $\mathbf{Q}_T = [\varphi(\mathbf{X}_q), \mathbf{Y}_{(q,t)} \sim \mathcal{N}(\mathbf{0}_q, \mathbf{I}_q)]$
3: $\mathbf{C}_T \subseteq \mathbf{Q}_T$             $\triangleright$ Random subset
4: **for** $t = T, \ldots, 1$ **do**
5:   $\mathbf{z} \sim \mathcal{N}(\mathbf{0}, \mathbf{I})$ if $t > 1$, else $\mathbf{z} = \mathbf{0}$
6:   $\mathbf{Y}_{(q,t-1)} = \frac{1}{\sqrt{\alpha_t}}\left(\mathbf{Y}_{(q,t)} - \frac{1-\alpha_t}{\sqrt{1-\bar{\alpha}_t}}\epsilon_\theta(\mathbf{C}_t, t, \mathbf{Q}_t)\right) + \sigma_t \mathbf{z}$
7:   $\mathbf{Q}_{t-1} = [\mathbf{M}_q, \mathbf{Y}_{(q,t-1)}]$
8:   $\mathbf{C}_{t-1} \subseteq \mathbf{Q}_{t-1}$     $\triangleright$ Same subset as in step 2
9: **end for**
10: **return** $f_0$ evaluated at coordinates $\varphi(\mathbf{X}_q)$

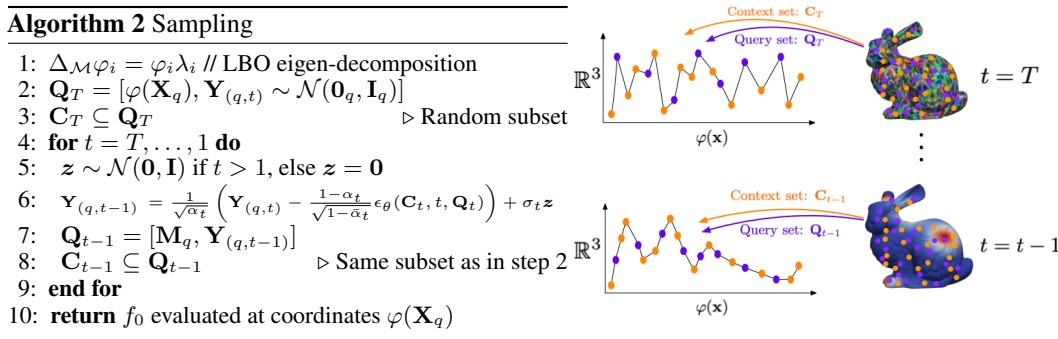

Figure 5: **Left:** MDF sampling algorithm. **Right**: Visual depiction of the sampling process for a field on the bunny manifold.

## 5 EXPERIMENTS

We validate the practicality of MDF via extensive experiments including synthetic and real-world problems. In Sect. 5.1 we provide results for learning distributions of functions on a fixed manifold (*e.g.* climate science), where functions change but manifolds are fixed across all functions. In addition, in Sect. 5.2 we show that MDF is robust to different manifold parametrizations. Finally, in Sect. 5.3 we also provide results on a generalized setting where manifolds are different for each function (*e.g.* molecule conformer generation). As opposed to generative models over images, we cannot rely on FID (Heusel et al., 2017) type metrics for evaluation since functions are defined on curved geometries. We borrow metrics from generative modeling of point cloud data (Achlioptas et al., 2018), namely Coverage (COV) and Minimum Matching Distance (MMD). We compute COV and MMD metrics based on the $l_2$ distance in signal space for corresponding vertices in the manifolds.

### 5.1 DISTRIBUTIONS OF FUNCTIONS ON A FIXED MANIFOLD

We evaluate MDF on 3 different manifolds that are fixed across functions: a sine wave, the Stanford bunny and a human mesh. These manifolds have an increasing average mean curvature $|K|$ (averaged over vertices), which serves as a measure for how distant they are from being globally Euclidean. On each manifold we define 3 function datasets: a Gaussian Mixture (GMM) with 3 components (where in each field the 3 components are randomly placed on the manifold), MNIST (LeCun et al., 1998) and CelebA-HQ (Karras et al., 2018) images. We use an off-the-shelf texture mapping approach (Sullivan & Kaszynski, 2019) to map images to manifolds, see Fig. 1. We compare MDF with Diffusion Probabilistic Fields (DPF) (Zhuang et al., 2023) a generative model for fields in ambient space, where points in the manifold are parametrized by the Fourier PE of its coordinates in 3D space. To provide a fair comparison we equate all the hyper-parameters in both MDF and DPF (Zhuang

|  | GMM | | MNIST | | CelebA-HQ | |
|---|---|---|---|---|---|---|
|  | COV↑ | MMD↓ | COV↑ | MMD↓ | COV↑ | MMD↓ |
| MDF | **0.444** | 0.01405 | **0.564** | **0.0954** | 0.354 | **0.11601** |
| DPF | 0.352 | **0.01339** | 0.552 | 0.09633 | **0.361** | 0.12288 |

Table 1: COV and MMD metrics for different datasets on the *wave* manifold (mean curvature $|K| = 0.004$).

|  | GMM | | MNIST | | CelebA-HQ | |
|---|---|---|---|---|---|---|
|  | COV↑ | MMD↓ | COV↑ | MMD↓ | COV↑ | MMD↓ |
| MDF | **0.575** | **0.00108** | **0.551** | **0.07205** | **0.346** | **0.11101** |
| DPF | 0.472 | 0.00120 | 0.454 | 0.11525 | 0.313 | 0.11530 |

Table 2: Results on the *bunny* manifold (mean curvature $|K| = 7.388$). As the mean curvature increases the boost of MDF over DPF (Zhuang et al., 2023) becomes larger across all datasets.

|  | GMM | | MNIST | | CelebA-HQ | |
|---|---|---|---|---|---|---|
|  | COV↑ | MMD↓ | COV↑ | MMD↓ | COV↑ | MMD↓ |
| MDF | **0.551** | **0.00100** | **0.529** | **0.08895** | **0.346** | **0.14162** |
| DPF | 0.479 | 0.00112 | 0.472 | 0.09537 | 0.318 | 0.14502 |

Table 3: *Human* manifold (mean curvature $|K| = 25.966$). At high mean curvatures MDF consistently outperforms DPF (Zhuang et al., 2023).

|  | $\mathcal{M} \to \mathcal{M}$ | | $\mathcal{M} \to \mathcal{M}_{\mathrm{iso}}$ | | $\mathcal{M}_{\mathrm{iso}} \to \mathcal{M}_{\mathrm{iso}}$ | |
|---|---|---|---|---|---|---|
|  | COV↑ | MMD↓ | COV↑ | MMD↓ | COV↑ | MMD↓ |
| MDF | **0.595** | **0.00177** | **0.595** | **0.00177** | **0.582** | **0.00191** |
| DPF | 0.547 | 0.00189 | 0.003 | 0.08813 | 0.306 | 0.00742 |

Table 4: Training MDF on a manifold $\mathcal{M}$ and evaluating it on an isometric transformation $\mathcal{M}_{\mathrm{iso}}$ does not impact performance, while being on par with training directly on the transformed manifold.

et al., 2023). Tab. 1-2-3 show results for the different approaches and tasks. We observe that MDF tends to outperform DPF (Zhuang et al., 2023), both in terms of covering the empirical distribution, resulting in higher COV, but also in the fidelity of the generated fields, obtaining a lower MMD. In particular, MDF outperforms DPF (Zhuang et al., 2023) across the board for manifolds of large mean curvature $|K|$. We attribute this behaviour to our choice of using intrinsic functional basis (*e.g.* eigen-functions of the LBO) to represent a coordinate system for points in the manifold. Fig. 1 shows a side to side comparison of real and generated functions on different manifolds obtained from MDF.

We also compare MDF with GASP (Dupont et al., 2022b), a generative model for continuous functions using an adversarial formulation. We compare MDF and GASP performance on the CelebA-HQ dataset (Karras et al., 2018) mapped on the bunny manifold. Additionally, we report results on the ERA5 climate dataset (Dupont et al., 2022b), which is composed of functions defined on the sphere $f : S^2 \to \mathbb{R}^1$ (see Fig. 1). For the ERA5 dataset we use spherical harmonics to compute $\varphi$, which are equivalent to the analytical eigen-functions of the LBO on the sphere (Lévy, 2006). To compare with GASP we use their pre-trained models to generate samples. In the case of CelebA-HQ, we use GASP to generate 2D images and map them to the bunny manifold using (Sullivan & Kaszynski, 2019). Experimental results in Tab. 5 show that MDF outperforms GASP in both ERA5(Hersbach et al., 2020) and CelebA-HQ datasets, obtaining both higher coverage but also higher fidelity in generated functions. This can be observed in Fig. 6 where the samples generated by MDF are visually crisper than those generated by GASP.

|  | ERA5 | | CelebA-HQ on $\mathcal{M}$ | |
|---|---|---|---|---|
|  | COV↑ | MMD↓ | COV↑ | MMD↓ |
| MDF | **0.347** | **0.00498** | **0.346** | **0.11101** |
| GASP | 0.114 | 0.00964 | 0.309 | 0.38979 |

Table 5: MDF outperforms GASP on ERA5(Hersbach et al., 2020) and CelebA-HQ both in terms of fidelity and distribution coverage. For GASP, we generate CelebA-HQ images and texture map them to the bunny manifold using (Sullivan & Kaszynski, 2019).

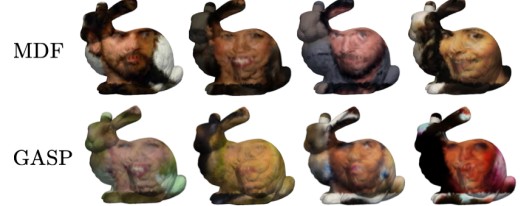

Figure 6: CelebA-HQ samples generated by MDF and GASP (Dupont et al., 2022b) on the bunny.

Furthermore, we ablate the performance of MDF as the number of eigen-functions used to compute the coordinate representation $\varphi$ increases (*e.g.* the eigen-decomposition of the LBO). For this task we use the bunny and the GMM dataset. Results in Fig. 7 show that performance initially increases with the number of eigen-functions up to a point where high frequency eigen-functions of the LBO are not needed to faithfully encode the distribution of functions.

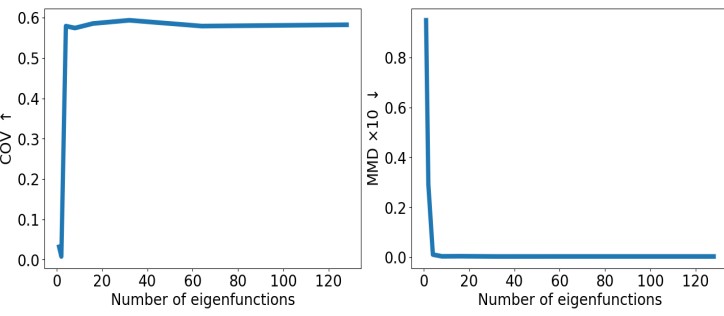

Figure 7: Performance of MDF as a function of the number of eigen-functions of the LBO, measured by COV and MMD metrics. As expected, performance increases initially as more eigen-functions are used, followed by a plateau phase for more than $k = 32$ eigen-functions.

## 5.2 MANIFOLD PARAMETRIZATION

MDF uses the eigen-functions of the LBO as positional embeddings. In practice, different real-world problems parametrize manifolds in different ways, and thus, have different ways of computing the LBO. For example, in computer graphics the usage of 3D meshes and cotangent Laplacians (Rustamov et al., 2007) is widespread. In computer vision, 3D geometry can also be represented as pointclouds which enjoy sparsity benefits and for which Laplacians can also be computed (Sharp & Crane, 2020). Finally, in computational chemistry problems, molecules are represented as undirected graphs of atoms connected by bonds, in this case graph Laplacians are commonly used (Maskey et al., 2022). In Fig. 8 we show the top-2 eigenvectors of these different Laplacians on the bunny manifold. In Tab. 6 we show the performance of MDF on the bunny mesh on different datasets using different manifold parametrizations and their respective Laplacian computation. These results show that MDF is relatively robust to different Laplacians and can be readily applied to any of these different settings by simply computing eigenvectors of the appropriate Laplacian.

|  | GMM | | MNIST | | CelebA-HQ | |
|---|---|---|---|---|---|---|
|  | COV↑ | MMD↓ | COV↑ | MMD↓ | COV↑ | MMD↓ |
| Graph | 0.575 | **0.00108** | 0.551 | 0.07205 | 0.346 | **0.11101** |
| Cotangent | 0.581 | 0.00384 | 0.568 | **0.06890** | **0.374** | 0.12440 |
| Pointcloud | **0.588** | 0.00417 | **0.571** | 0.06909 | 0.337 | 0.12297 |

Table 6: Performance of MDF using different Laplacians for different datastets on the *bunny* manifold, where we see that MDF is relatively robust and can be readily deployed on different settings depending on the manifold parametrization.

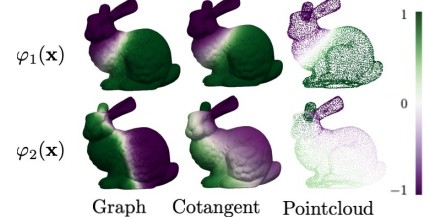

Figure 8: Visualizing top-2 eigenvectors on the bunny manifold for Graph, Cotangent and Pointcloud (Sharp & Crane, 2020) Laplacians.

## 5.3 GENERALIZING ACROSS MANIFOLDS

We now generalize the problem setting to learning distributions over functions where each function is defined on a different manifold. In this setting, the training set is defined as $\{f_i\}_{i=0:N}$ with functions $f_i : \mathcal{M}_i \to \mathcal{Y}$ mapping elements from different manifolds $\mathcal{M}_i$ to a shared signal space $\mathcal{Y}$. This is a generalization of the setting in Sect. 5.1 where functions are defined as $f_i : \mathcal{M} \to Y$, with the manifold $\mathcal{M}$ being *fixed* across $f_i$'s. This generalized setting is far more complex than the *fixed* setting since the model not only has to figure out the distribution of functions but also it needs to represent different manifolds in a consistent manner. To evaluate the performance of MDF in this setting we tackle the challenging problem of molecule conformer generation (Xu et al., 2021; 2022; Ganea et al., 2021; Jing et al., 2022) which is a fundamental task in computational chemistry and requires models to handle multiple manifolds. In this problem, manifolds $\mathcal{M}_i$ are parametrized as graphs that encode the connectivity structure between atoms of different types. From MDF's perspective a conformer is then a function $f_i : \mathcal{M}_i \to \mathbb{R}^3$ that maps elements in the graph (*e.g.* atoms) to a point in 3D space. Note that graphs are just one of different the manifold representations that are amenable for MDF as show in Sect. 5.2.

Following the standard setting for molecule conformer prediction we use the GEOM-QM9 dataset (Ruddigkeit et al., 2012; Ramakrishnan et al., 2014) which contains $\sim 130K$ molecules ranging from $\sim 10$ to $\sim 40$ atoms. We report our results in Tab. 7 and compare with CGCF (Xu et al., 2021), GeoDiff (Xu et al., 2022), GeoMol (Ganea et al., 2021) and Torsional Diffusion (Jing et al., 2022). Note that both GeoMol (Ganea et al., 2021) and Torsional Diffusion (Jing et al., 2022) make strong assumptions about the geometric structure of molecules and model domain-specific characteristics like torsional angles of bonds. In contraposition, MDF simply models the distribution of 3D coordinates of atoms without making any assumptions about the underlying structure. We use the same train/val/test splits as Torsional Diffusion (Jing et al., 2022) and use the same metrics to compare the generated and ground truth conformer ensembles: Average Minimum RMSD (AMR) and Coverage. These metrics are reported both for precision, measuring the accuracy of the generated conformers, and recall, which measures how well the generated ensemble covers the ground-truth ensemble. We generate 2K conformers for a molecule with K ground truth conformers. Note that in this setting, models are evaluated on unseen molecules (*e.g.* unseen manifolds $\mathcal{M}_i$).

We report results on Tab. 7 where we see how MDF outperforms previous approaches. It is important to note that MDF is a general approach for learning functions on manifolds that does not make any assumptions about the intrinsic geometric factors important in conformers like torsional angles in Torsional Diffusion (Jing et al., 2022). This makes MDF simpler to implement and applicable to other settings in which intrinsic geometric factors are not known.

| | Recall | | | | Precision | | | |
|---|---|---|---|---|---|---|---|---|
| | Coverage ↑ | | AMR ↓ | | Coverage ↑ | | AMR ↓ | |
| | mean | median | mean | median | mean | median | mean | median |
| CGCF | 69.47 | 96.15 | 0.425 | 0.374 | 38.20 | 33.33 | 0.711 | 0.695 |
| GeoDiff | 76.50 | 100.00 | 0.297 | 0.229 | 50.00 | 33.50 | 0.524 | 0.510 |
| GeoMol | 91.50 | 100.00 | 0.225 | 0.193 | 87.60 | 100.00 | 0.270 | 0.241 |
| Torsional Diff. | 92.80 | 100.00 | 0.178 | 0.147 | **92.70** | 100.00 | 0.221 | 0.195 |
| MDF (ours) | **95.30** | 100.00 | **0.124** | **0.074** | 91.50 | 100.00 | **0.169** | **0.101** |
| MDF ($k = 16$) | **94.87** | 100.00 | **0.139** | 0.093 | **87.54** | 100.00 | 0.220 | 0.151 |
| MDF ($k = 8$) | 94.28 | 100.00 | 0.162 | 0.109 | 84.27 | 100.00 | 0.261 | 0.208 |
| MDF ($k = 4$) | 94.57 | 100.00 | 0.145 | 0.093 | 86.83 | 100.00 | 0.225 | 0.151 |
| MDF ($k = 2$) | 93.15 | 100.00 | 0.152 | **0.088** | 86.97 | 100.00 | **0.211** | **0.138** |

Table 7: Molecule conformer generation results for GEOM-QM9 dataset. MDF obtains comparable or better results than the state-of-the-art Torsional Diffusion (Jing et al., 2022), without making any explicit assumptions about the geometric structure of molecules (*i.e.* without modeling torsional angles). In addition, we show how performance of MDF changes as a function of the number of eigen-functions $k$. Interestingly, with as few as $k = 2$ eigen-functions MDF is able to generate consistent accurate conformations.

Finally, In the appendix we present additional results that carefully ablate different architectures for the score network $\epsilon_\theta$ in A.7.1. As well as an extensive study of the robustness of MDF to both rigid and isometric transformations of the manifold $\mathcal{M}$ A.7.2. Finally, we also show conditional inference results on the challenging problem of PDEs on manifolds A.7.3.

## 6 CONCLUSIONS

In this paper we introduced MDF a diffusion probabilistic model that is capable of capturing distributions of functions defined on general Riemannian manifolds. We leveraged tools from spectral geometry analysis and use the eigen-functions of the manifold Laplace-Beltrami Operator to define an intrinsic coordinate system on which functions are defined. This allows us to design an efficient recipe for training a diffusion probabilistic model of functions whose domain are arbitrary geometries. Our results show that we can capture distributions of functions on manifolds of increasing complexity outperforming previous approaches, while also enabling the applications of powerful generative priors to fundamental scientific problems like forward and inverse solutions to PDEs, climate modeling, and molecular chemistry.

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

# A  APPENDIX

## A.1  BROADER IMPACT STATEMENT

When examining the societal implications of generative models, certain critical elements warrant close attention. These include the potential misuse of generative models to fabricate deceptive data, such as "DeepFakes" (Mirsky & Lee, 2021), the risk of training data leakage and associated privacy concerns (Tinsley et al., 2021), and the potential to amplify existing biases in the training data (Jain et al., 2020). For a comprehensive discussion on ethical aspects in the context of generative modeling, readers are directed to (Rostamzadeh et al., 2021).

## A.2  LIMITATIONS AND FUTURE WORK

As MDF advances in learning function distributions over Riemannian manifolds, it does encounter certain constraints and potential areas of future enhancement. One primary challenge is the computational demand of the transformer-based score network in its basic form, even at lower resolutions. This stems from the quadratic cost of calculating attention over context and query pairs. To mitigate this, the PerceiverIO architecture, which scales in a linear manner with the number of query and context points, is utilized (Jaegle et al., 2022) in our experiments. Further exploration of other efficient transformer architectures could be a promising direction for future work (Zhai et al., 2022; Dao et al., 2022). Furthermore, MDF, much like DDPM (Ho et al., 2020), iterates over all time steps during sampling to generate a field during inference, a process slower than that of GANs. Current studies have accelerated sampling (Song et al., 2021a), but at the expense of sample quality and diversity. However, it's worth noting that improved inference methods such as (Song et al., 2021a) can be seamlessly incorporated into MDF.

Since MDF has the capability to learn distributions over fields defined on various Riemannian manifolds within a single model, in future work we are poised to enhance its capacity for comprehending and adapting to a broader range of geometrical contexts. This adaptability will further pave the way towards the development of general foundation models to scientific and engineering challenges, which can better account for the intricate geometric intricacies inherent in real-world scenarios.

For example, we aim to extend the application of MDF to inverse problems in PDEs. A noteworthy attribute of our model is its inherent capability to model PDEs on Riemannian manifolds trivially. The intrinsic structure of MDF facilitates not only the understanding and solving of forward problems, where PDEs are known and solutions to the forward problem are needed, but also inverse problems, where certain outcome or boundary conditions are known and the task is to determine the underlying PDE. Expanding our application to handle inverse problems in PDEs on Riemannian manifolds can have profound implications for complex systems modeling, as it enhances our understanding of the manifold structures and the way systems governed by PDEs interact with them.

## A.3  DISCUSSION ON COMPUTING EMBEDDINGS

When considering how to compute embeddings for points in a manifold $\mathcal{M}$ there are several options to explore. The simplest one is to adopt the ambient space in which the manifold is embedded as a coordinate system to represent points (eg. a plain coordinate approach). For example, in the case of 3D meshes one can assign a coordinate in $\mathbb{R}^3$ to every point in the mesh. As shown in Tab. 1-2-3-4 this approach (used by DPF) is outperformed by MDF. In addition, in Sect. A.6.2 we show that this approach is not robust wrt rigid or isometric transformations of the manifold. Note that manifolds are not always embedded in an ambient space. For example, in molecular conformation, molecular graphs only represent connectivity structure between atoms but are not necessarily embedded in a higher dimensional space.

Another method that one can consider is to use a local chart approach. Local charts are interesting because they provide a way assigning a set of coordinates to points in a local region of the manifold. While the manifold may have arbitrary curvature, local charts are always Euclidean spaces. Each point in the manifold can be described by a unique set of coordinates in the chart, but different charts may overlap. However, this requires computing transformations (often complex to implement) to convert coordinates from one chart to another.

Finally, the eigen-functions of the LBO not only provide a way of assigning a coordinate to points on a manifold but also do this by defining an intrinsic coordinate system. This intrinsic coordinate system is global, and does not require transformations like local charts do. In addition, this intrinsic coordinate system is robust wrt rigid or isometric transformations of the manifold (ref A.6.2). Summarizing, this intrinsic coordinate system is a more fundamental way of describing the manifold, based on its own inherent properties, without reference to an external ambient space.

### A.4 IMPLEMENTATION DETAILS

In this section we describe implementation details for all our experiments. These include all details about the data: manifolds and functions, as well as details for computing the eigen-functions $\varphi$. We also provide hyper-parameters and settings for the implementation of the score field network $\epsilon_\theta$ and compute used for each experiment in the paper.

#### A.4.1 DATA

Unless explicitly described in the main paper, we report experiments on 5 different manifolds which we show in Fig. 9. This manifolds are: a parametric sine wave Fig. 9(a) computed using (Sullivan & Kaszynski, 2019) containing 1024 vertices. The Stanford bunny with 5299 vertices Fig. 9(b). A human body mesh from the Tosca dataset (Bronstein et al., 2008) containing 4823 vertices, show in Fig. 9(c). A cat mesh and its reposed version from (Sumner & Popovic, 2004), show in Fig. 9(d) and Fig. 9(e), respectively containing 7207 vertices. To compute the mean curvature values $|K|$ for each mesh reported in the main paper we compute the absolute value of the average mean curvature, which we obtain using (Sullivan & Kaszynski, 2019).

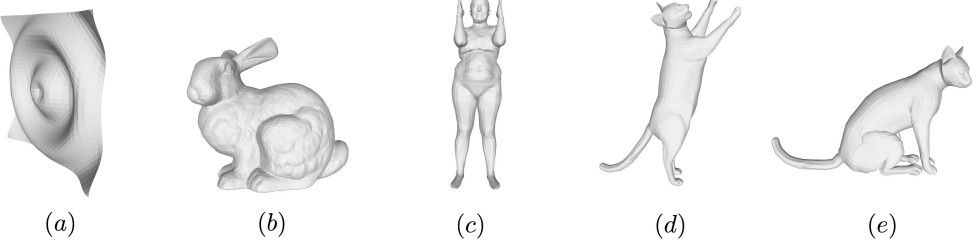

$(a)$      $(b)$      $(c)$      $(d)$      $(e)$

Figure 9: Manifolds used in the different experiments throughout the paper. (a) Wave. (b) Bunny. (c) Human (Bronstein et al., 2008). (d) Cat (Sumner & Popovic, 2004). (e) Cat (re-posed) (Sumner & Popovic, 2004).

In terms of datasets of functions on these manifolds we use the following:

- A Gaussian Mixture Model (GMM) dataset with 3 components, where in each field the 3 components are randomly placed on the specific manifold. We define a held out test set containing 10k samples.

- MNIST (LeCun et al., 1998) and CelebA-HQ (Karras et al., 2018) datasets, where images are texture mapped into the meshes using (Sullivan & Kaszynski, 2019), models are evaluated on the standard tests sets for these datasets.

- The ERA5 (Hersbach et al., 2020) dataset used to compare with GASP (Dupont et al., 2022b) is available at https://github.com/EmilienDupont/neural-function-distributions. This dataset contains a train set of 8510 samples and a test set of 2420 samples, which are the settings used in GASP (Dupont et al., 2022b). To compare with GASP we used their pretrained model available available at [1]

---

[1]https://github.com/EmilienDupont/neural-function-distributions

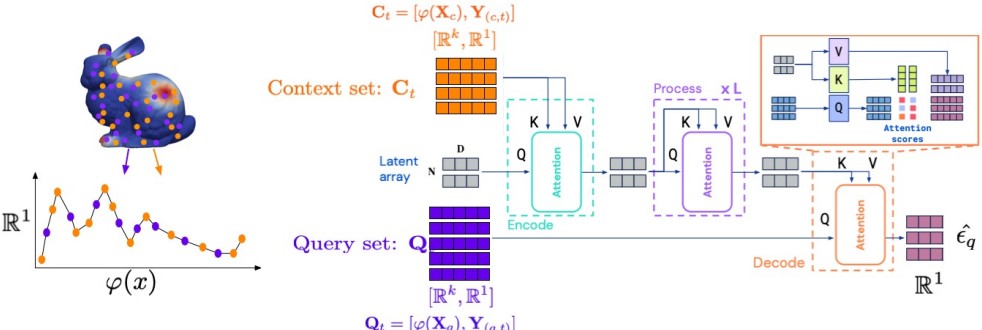

Figure 10: Interaction between context and query pairs in the PerceiverIO architecture. Context pairs $\mathbf{C}_t$ attend to a latent array of learnable parameters via cross attention. The latent array then goes through several self attention blocks. Finally, the query pairs $\mathbf{Q}_t$ cross-attend to the latent array to produce the final noise prediction $\hat{\epsilon}_q$.

## A.5 COMPUTING THE LAPLACIAN AND $\varphi$

In practice, for general geometries (*e.g.* general 3D meshes with $n$ vertices) we compute eigenvectors of the symmetric normalized graph Laplacian $\mathbf{L}$. We define $\mathbf{L}$ as follows:

$$\mathbf{L} = \mathbf{D}^{-\frac{1}{2}}(\mathbf{D} - \mathbf{A})\mathbf{D}^{-\frac{1}{2}}, \tag{3}$$

where $\mathbf{A} \in \{0, 1\}^{n \times n}$ is the discrete adjacency matrix and $\mathbf{D}$ is the diagonal degree matrix of the mesh graph. Note that eigenvectors of $\mathbf{L}$ converge to the eigen-functions of the LBO $\Delta_{\mathcal{M}}$ as $n \to \infty$ (Belkin & Niyogi, 2001; Bengio et al., 2003b;a). The eigen-decomposition of $\mathbf{L}$ can be computed efficiently using sparse eigen-problem solvers (Hernandez et al., 2009) and only needs to be computed once during training. Note that eigen-vectors of $\mathbf{L}$ are only defined for the mesh vertices. In MDF, we sample random points on the mesh during training and interpolate the eigenvector representation $\varphi$ of the vertices in the corresponding triangle using barycentric interpolation.

### A.5.1 SCORE FIELD NETWORK $\epsilon_\theta$

In MDF, the score field's design space covers all architectures that can process irregularly sampled data, such as Transformers (Vaswani et al., 2017) and MLPs (Tolstikhin et al., 2021). The model is primarily implemented using PerceiverIO (Jaegle et al., 2022), an effective transformer architecture that encodes and decodes. The PerceiverIO was chosen due to its efficiency in managing large numbers of elements in the context and query sets, as well as its natural ability to encode interactions between these sets using attention. Figure 10 demonstrates how these sets are used within the PerceiverIO architecture. To elaborate, the encoder maps the context set into latent arrays (i.e., a group of learnable vectors) through a cross-attention layer, while the decoder does the same for query set. For a more detailed analysis of the PerceiverIO architecture refer to (Jaegle et al., 2022).

The time-step $t$ is incorporated into the score computation by concatenating a positional embedding representation of $t$ to the context and query sets. The specific PerceiverIO settings used in all quantitatively evaluated experiments are presented in Tab.8. Practically, the MDF network consists of 12 transformer blocks, each containing 1 cross-attention layer and 2 self-attention layers, except for GEOM-QM9 we use smaller model with 6 blocks. Each of these layers has 4 attention heads. Fourier position embedding is used to represent time-steps $t$ with 64 frequencies. An Adam (Kingma & Ba, 2015) optimizer is employed during training with a learning rate of $1e - 4$. We use EMA with a decay of 0.9999. A modified version of the publicly available repository is used for PerceiverIO [2].

---

[2] https://huggingface.co/docs/transformers/model_doc/perceiver

| Hyper-parameter | Wave | Bunny | Human | GEOM-QM9 |
|---|---|---|---|---|
| train res. | 1024 | 5299 | 4823 | variable |
| #context set | 1024 | 5299 | 4823 | variable |
| #query set | 1024 | 5299 | 4823 | variable |
| #eigenfuncs $(k)$ | 64 | 64 | 64 | 28 |
| #freq pos. embed $t$ | 64 | 64 | 64 | 64 |
| #latents | 1024 | 1024 | 1024 | 512 |
| #dim latents | 512 | 512 | 512 | 256 |
| #blocks | 12 | 12 | 12 | 6 |
| #dec blocks | 1 | 1 | 1 | 1 |
| #self attends per block | 2 | 2 | 2 | 2 |
| #self attention heads | 4 | 4 | 4 | 4 |
| #cross attention heads | 4 | 4 | 4 | 4 |
| batch size | 96 | 96 | 96 | 96 |
| lr | $1e-4$ | $1e-4$ | $1e-4$ | $1e-4$ |
| epochs | 1000 | 1000 | 1000 | 250 |

Table 8: Hyperparameters and settings for MDF on different manifolds.

### A.5.2 COMPUTE

Each model was trained on an machine with 8 Nvidia A100 GPUs, we trained models for 3 days.

### A.6 METRICS

Instead of using FID type metrics commonly used for generative models over images (Heusel et al., 2017), we must take a different approach for evaluating functions on curved geometries. Our suggestion is to use metrics from the field of generative modeling of point cloud data (Achlioptas et al., 2018), specifically Coverage (COV) and Minimum Matching Distance (MMD).

- **Coverage** (COV) refers to how well the generated data set represents the test set. We first identify the closest neighbour in the generated set for each field in the test set. COV is then calculated as the proportion of fields in the generated set that have corresponding fields in the test set. The distance between fields is determined using the average $l_2$ distance in signal space on the vertices of the mesh, usually in either $\mathbb{R}^1$ or $\mathbb{R}^3$ space in our experiments. A high COV score implies that the generated samples adequately represent the real samples.

- **Minimum Matching Distance** (MMD), on the other hand, provides a measure of how accurately the fields are represented in the test set. This measure is required because in the COV metric matches don't necessarily have to be close. To gauge the fidelity of the generated fields against the real ones, we pair each field in the generated set with its closes neigbour in the test set (MMD), averaging these distances for our final result. This process also utilizes the $l_2$ distance in signal space on the mesh vertices. MMD provides a good correlation with the authenticity of the generated set, as it directly depends on the matching distances.

As a summary, COV and MMD metrics are complementary to each other. A model captures the distribution of real fields with good fidelity when MMD is small and COV is large. In particular, at equivalent levels of MMD a higher COV is desired (Achlioptas et al., 2018), and vice-versa. This observation correlates well with our results shown in Tab. 1-2-3 on the main paper, where MDF obtains comparable or better MMD score that DPF (Zhuang et al., 2023) while greatly improving COV.

### A.7 ADDITIONAL EXPERIMENTS

In this section we provide additional empirical results using different network architectures to implement the score field network $\epsilon_\theta$. Furthermore, we provide additional experiments on robustness to discretization.

### A.7.1 ARCHITECTURE ABLATION

The construction of MDF does not rely on a specific implementation of the score network $\epsilon_\theta$. The score model's design space encompasses a broad range of options, including all architectures capable of handling irregular data like transformers or MLPs. To substantiate this, we conducted an evaluation on the GMM dataset and the Stanford bunny at a resolution of 602 vertices, comparing three distinct architectures: a PerceiverIO (Jaegle et al., 2022), a standard Transformer Encoder-Decoder (Vaswani et al., 2017), and an MLP-mixer (Tolstikhin et al., 2021). For a fair comparison, we approximated the same number of parameters (around 55M) and settings (such as the number of blocks, parameters per block, etc.) for each model and trained them over 500 epochs. Note that because of these reasons the numbers reported in this section are not directly comparable to those shown in the main paper. We simplified the evaluation by using an equal number of context and query pairs. Both the Transformer Encoder and MLP-mixer process context pairs using their respective architectures; the resulting latents are then merged with corresponding query pairs and fed into a linear projection layer for final prediction.

In Tab. 9 we show that the MDF formulation is compatible with different architectural implementations of the score field model. We observe relatively uniform performance across various architectures, ranging from transformer-based to MLPs. Similar patterns are noted when examining qualitative samples displayed in Fig. 11, corroborating our assertion that MDF's advantages stem from its formulation rather than the specific implementation of the score field model. Each architecture brings its own strengths—for instance, MLP-mixers enable high throughput, transformer encoders are easy to implement, and PerceiverIO facilitates the handling of large and variable numbers of context and query pairs. We posit that marrying the strengths of these diverse architectures promises substantial advancement for MDF. Please note, these empirical results aren't directly comparable to those reported elsewhere in the paper, as these models generally possess around $50\%$ of the parameters of the models used in other sections.

| | COV ↑ | MMD ↓ |
|---|---|---|
| PeceiverIO (Jaegle et al., 2022) | 0.569 | 0.00843 |
| Transf. Enc-Dec (Vaswani et al., 2017) | 0.581 | 0.00286 |
| MLP-mixer (Tolstikhin et al., 2021) | 0.565 | 0.00309 |

Table 9: **Quantitative evaluation of image generation** on the GMM + Stanford Bunny dataset for different implementations of the score field $\epsilon_\theta$.

Finally, to measure the effect of random training seed for weight initialization we ran the exact same model fixing all hyper-parameters and training settings. For this experiment we used the PerceiverIO architecture and the GMM dataset on the Stanford bunny geometry with 602 vertices. We ran the same experiment 3 times and measured performance using COV and MMD metrics. Our results show that across the different training runs MDF obtained COV=$0.569 \pm 0.007$ and MMD=$0.00843 \pm 0.00372$.

### A.7.2 ROBUSTNESS OF MDF

We evaluate MDF's robustness to rigid and isometric transformations of the training manifold $\mathcal{M}$. We use the *cat* category geometries from (Sumner & Popovic, 2004) and build a dataset of different fields on the manifold by generating 2 gaussians around the area of the tail and the right paw of the cat. Note that every field is different since the gaussians are centered at different points in the tail and right paw, see Fig. 13(a). During training, the model only has access to fields defined on a fixed manifold $\mathcal{M}$ (see Fig. 13(a)). We then evaluate the model on either a rigid $\mathcal{M}_{\text{rigid}}$ (shown in Fig. 13(b)) or isometric $\mathcal{M}_{\text{iso}}$ (Fig. 13(c)) transformation of $\mathcal{M}$. Qualitatively comparing the transfer results of MDF with DPF (Zhuang et al., 2023) in Fig. 13(d)-(e), we see a marked difference in fidelity and coverage of the distribution.

In Fig. 13 we show how performance changes as the magnitude of a rigid transformation (*e.g.* a rotation about the $z-$axis) of $\mathcal{M}$ increases. As expected, the performance of DPF (Zhuang et al., 2023) sharply declines as we move away from the training setting, denoted by a rotation of 0 radians. However, MDF obtains a stable performance across transformations, this is due to the eigen-function basis being intrinsic to $\mathcal{M}$, and thus, invariant to rigid transformations. In addition, in Tab. 4 we show results under an isometric transformation of the manifold (*e.g.* changing the pose of the cat, see Fig.

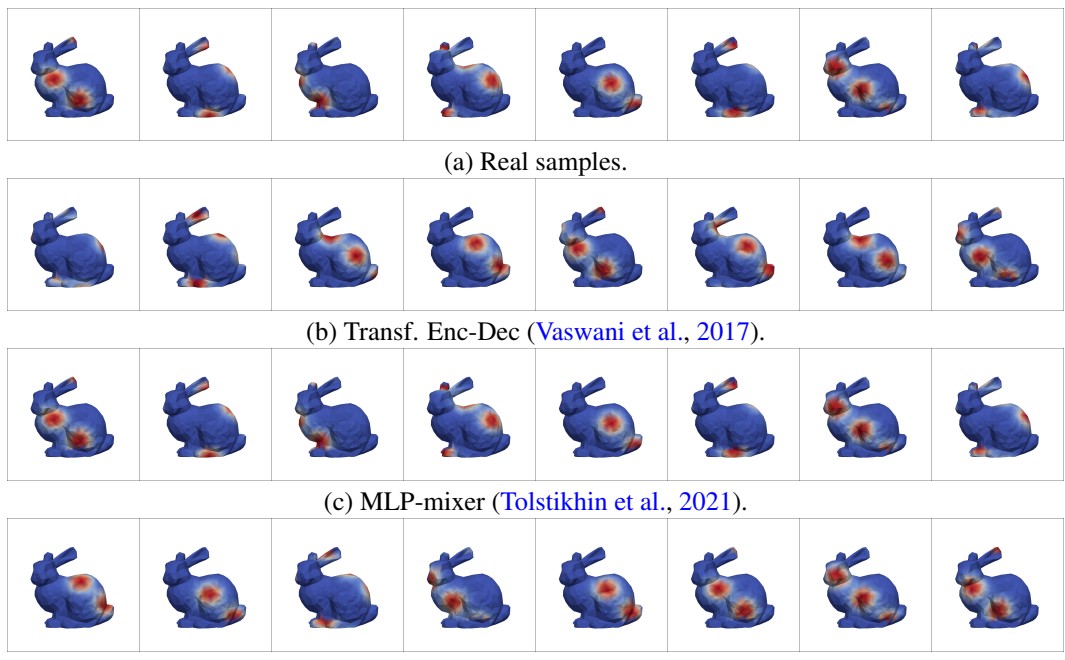

(a) Real samples.

(b) Transf. Enc-Dec (Vaswani et al., 2017).

(c) MLP-mixer (Tolstikhin et al., 2021).

(d) PerceiverIO (Jaegle et al., 2022).

Figure 11: **Qualitative comparison** of different architectures to implement the score field model $\epsilon_\theta$.

13(c)). As in the rigid setting, the performance of DPF (Zhuang et al., 2023) sharply declines under an isometric transformation while MDF keeps performance constant. In addition, transferring to an isometric transformation ($\mathcal{M} \to \mathcal{M}_{\text{iso}}$) performs comparably with directly training on the isometric transformation ($\mathcal{M}_{\text{iso}} \to \mathcal{M}_{\text{iso}}$) up to small differences due to random weight initialization.

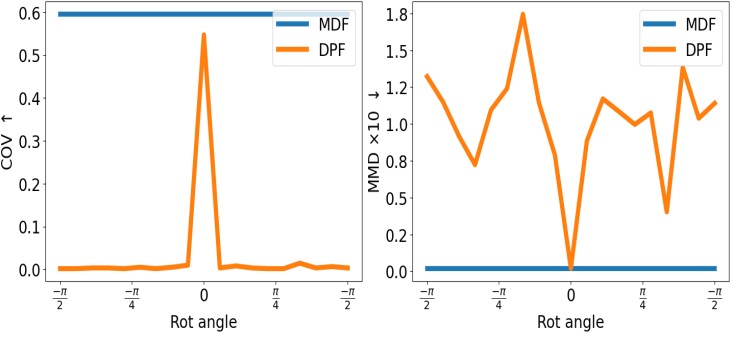

Figure 12: Robustness of MDF and DPF (Zhuang et al., 2023) with respect to rigid transformations of $\mathcal{M}$. The distribution of fields learned by MDF is invariant with respect to rigid transformations, while DPF (Zhuang et al., 2023) collapses due to learning distributions in ambient space.

We also provide transfer results to different discretizations of $\mathcal{M}$. To do so, we train MDF on a low resolution discretization of a manifold and evaluate transfer to a high resolution discretization. We use the GMM dataset and the bunny manifold at 2 different resolutions: 1394 and 5570 vertices, which we get by applying loop subdivision (Loop, 1987) to the lowest resolution mesh. Theoretically, the Laplacian eigenvectors $\varphi$ are only unique up to sign, which can result in ambiguity when transferring a pre-trained model to a different discretization. Empirically we did not find this to be an issue in our experiments. We hypothesize that transferring MDF from low to high resolution discretizations is largely a function of the number of eigen-functions used to compute $\varphi$. This is because eigenfunctions of small eigenvalue represent low-frequency components of the manifold which are more stable across different discretizations. In Fig. 14 we report transfer performance as a function of the number of eigen-functions used to compute $\varphi$. We observe an initial regime where more eigenfunctions aid in transferring (up to 64 eigen-functions) followed by a stage where high-frequency eigen-functions negatively impact transfer performance.

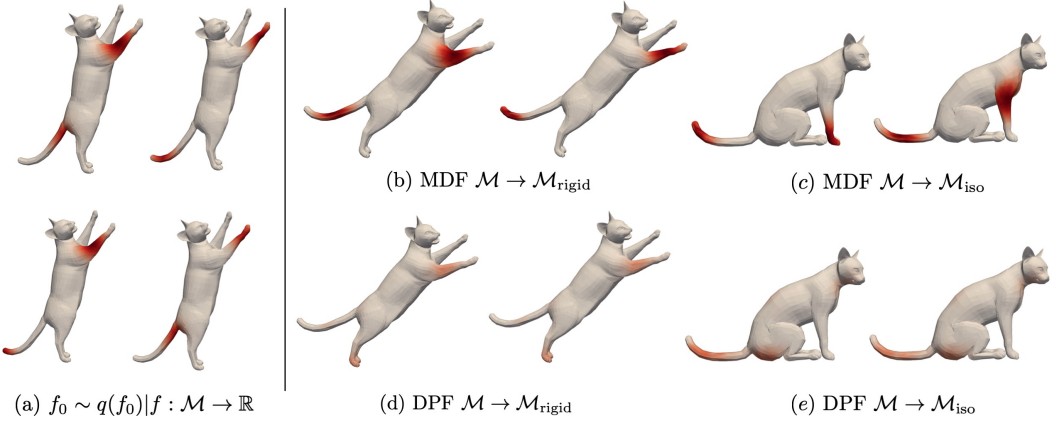

(b) MDF $\mathcal{M} \to \mathcal{M}_{\text{rigid}}$      (c) MDF $\mathcal{M} \to \mathcal{M}_{\text{iso}}$

(a) $f_0 \sim q(f_0)|f : \mathcal{M} \to \mathbb{R}$     (d) DPF $\mathcal{M} \to \mathcal{M}_{\text{rigid}}$     (e) DPF $\mathcal{M} \to \mathcal{M}_{\text{iso}}$

Figure 13: (a) **Training set** composed of different fields $f : \mathcal{M} \to \mathbb{R}$ where 2 gaussians are randomly placed in the tail and the right paw of the cat. Fields generated by **transferring** the MDF pre-trained on $\mathcal{M}$ to (b) a rigid and (c) an isometric transformation of $\mathcal{M}$. Fields generated by **transferring** the DPF (Zhuang et al., 2023) pre-trained on $\mathcal{M}$ to (d) a rigid and (e) an isometric transformation of $\mathcal{M}$.

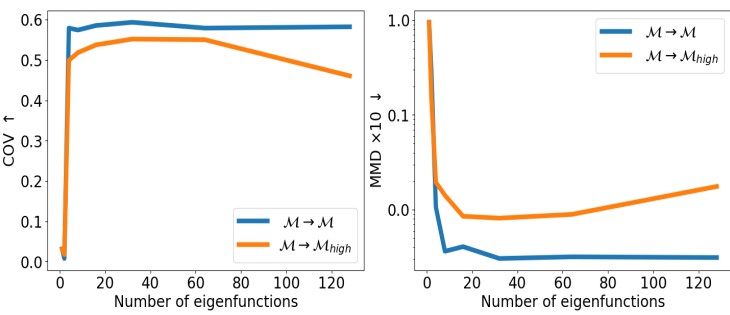

Figure 14: Transferring MDF from low to high resolution discretizations as a function of the number of eigenfunctions. We observe that eigen-functions of small eigenvalue transfer better since they encode coarse (*i.e.* low-frequency) information of the manifold.

We additionally run a transfer experiment between low resolution and high resolution discretizations of a different manifold (*e.g.* a mesh of the letter 'A', show in Fig. 15(b)). In this setting the low resolution mesh contains 1000 vertices and the high resolution mesh contains 4000 vertices. As show in Fig. 16 the results are consistent across manifolds, and a similar trend as in Fig. 14 can be observed. This trend further reinforces our hypothesis that low frequency eigen-functions transfer better across discretization than high frequency ones.

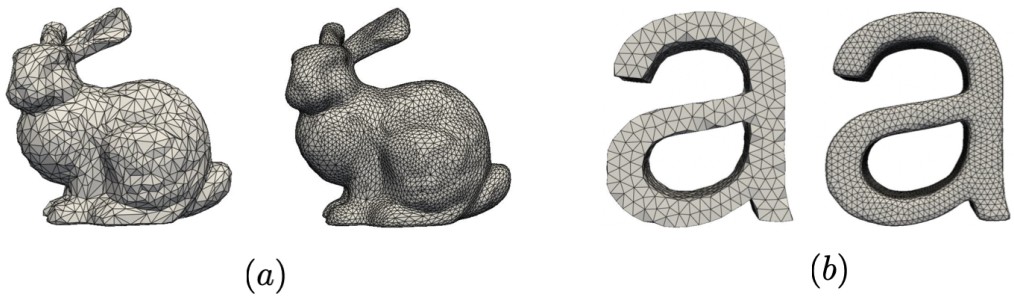

$(a)$                 $(b)$

Figure 15: (a) Low and high resolution discretizations of the Stanford bunny manifold used for the transfer experiments in the main paper (Fig. 14). (b) Low and high resolution discretizations of the letter 'A' manifold, used for the experiments in Fig. 16.

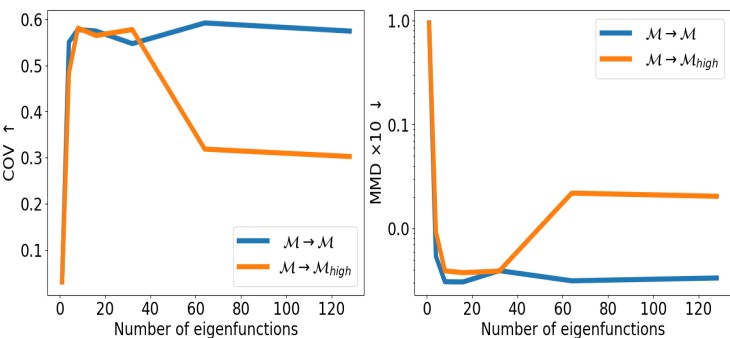

Figure 16: Transferring MDF from a less detailed to a more detailed discretization depends on the number of eigen-functions. It's noteworthy that eigen-functions with small eigenvalues have better transferability as they represent the broad, or low-frequency, information of the manifold.

### A.7.3 CONDITIONAL INFERENCE ON PDEs

In this section we evaluate MDF on conditional inference tasks. In particular, we create a dataset of different simulations of the heat diffusion PDE on a manifold. As a result, every sample in our training distribution $f_0 \sim q(f_0)$ is a temporal field $f : \mathcal{M} \times \mathbb{R} \to \mathbb{R}$. We create a training set of 10k samples where each sample is a rollout of the PDE for 10 steps given initial conditions. We generate initial conditions by uniformly sampling 3 gaussian heat sources of equivalent magnitude on the manifold and use FEM (Reddy, 2019) to compute the rollout over time. For this experiment we use a version of the bunny mesh with 602 vertices as a manifold $\mathcal{M}$ and set the diffusivity term of the PDE to $D = 0.78$. We then train MDF on this training set of temporal fields $f : \mathcal{M} \times \mathbb{R} \to \mathbb{R}$, which in practice simply means concatenating a Fourier PE of the time step to the eigen-functions of the LBO.

We tackle the forward problem where we are given the initial conditions of the PDE and the model is tasked to predict the forward dynamics on a test set of 60 held out samples. To perform conditional inference with MDF we follow the recipe in (Lugmayr et al., 2022) which has been successful in the image domain. We show the forward dynamics predicted by FEM (Reddy, 2019) on Fig. 17(a) and MDF Fig. 17(b) for the same initial conditions in the held out set. We see how MDF successfully captures temporal dynamics, generating a temporal field consistent with observed initial conditions. Evaluating the full test set MDF obtains an mean squared error MSE $= 4.77e10 - 3$. In addition, MDF can directly be used for inverse problems (Isakov, 2006). Here we focus on inverting the full dynamics of the PDE, conditioned on sparse observations. Fig. 17(c) shows sparse observations of the FEM rollout, amounting to observing 10% of the field. Fig. 17(d) shows a probabilistic solution to the inverse PDE problem generated by MDF which is consistent with the FEM dynamics in Fig. 17(a).

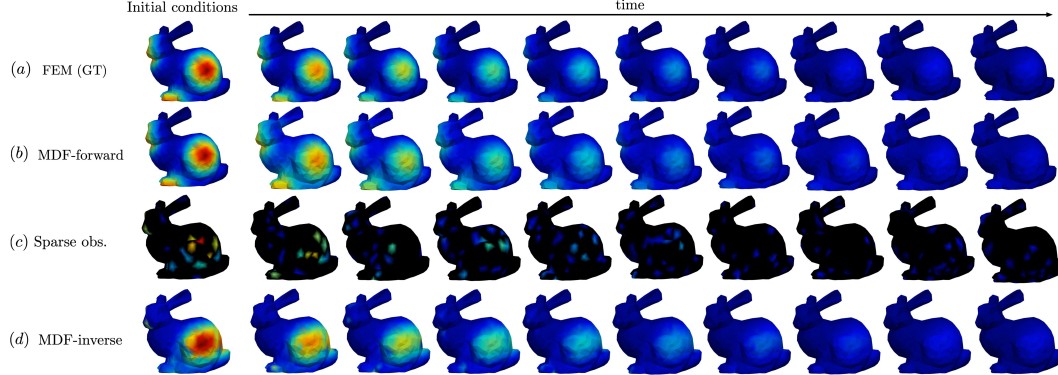

Figure 17: (a) Forward prediction of the heat diffusion PDE computed with FEM (Reddy, 2019). (b) Conditionally sampled field generated by MDF. (c) Sparse observations of the FEM solution for inverse prediction. (d) Conditionally sampled inverse solution generated by MDF.

## A.8 ADDITIONAL VISUALIZATIONS

In this section we provide additional visualizations of experiments in the main paper. We show real and generated fields for the wave manifold (Fig. 18), ERA5 dataset (Hersbach et al., 2020) (Fig. 19) and GMM dataset on the bunny (Fig. 20) and human (Bronstein et al., 2008) manifolds (see Fig. 21). In summary, MDF captures the distribution of real fields for different datasets and manifolds, with high fidelity and coverage.

Finally, under `./videos` we include two video visualizations:

- A visualization of training data as well as the sampling process for the MNIST dataset on the wave manifold.
- A visualization of GT and temporal fields generated by MDF for the PDE dataset introduced in Sect. A.7.3.
- A visualization of the sampling process for QM9 molecules for experiments Sect. 5.3.

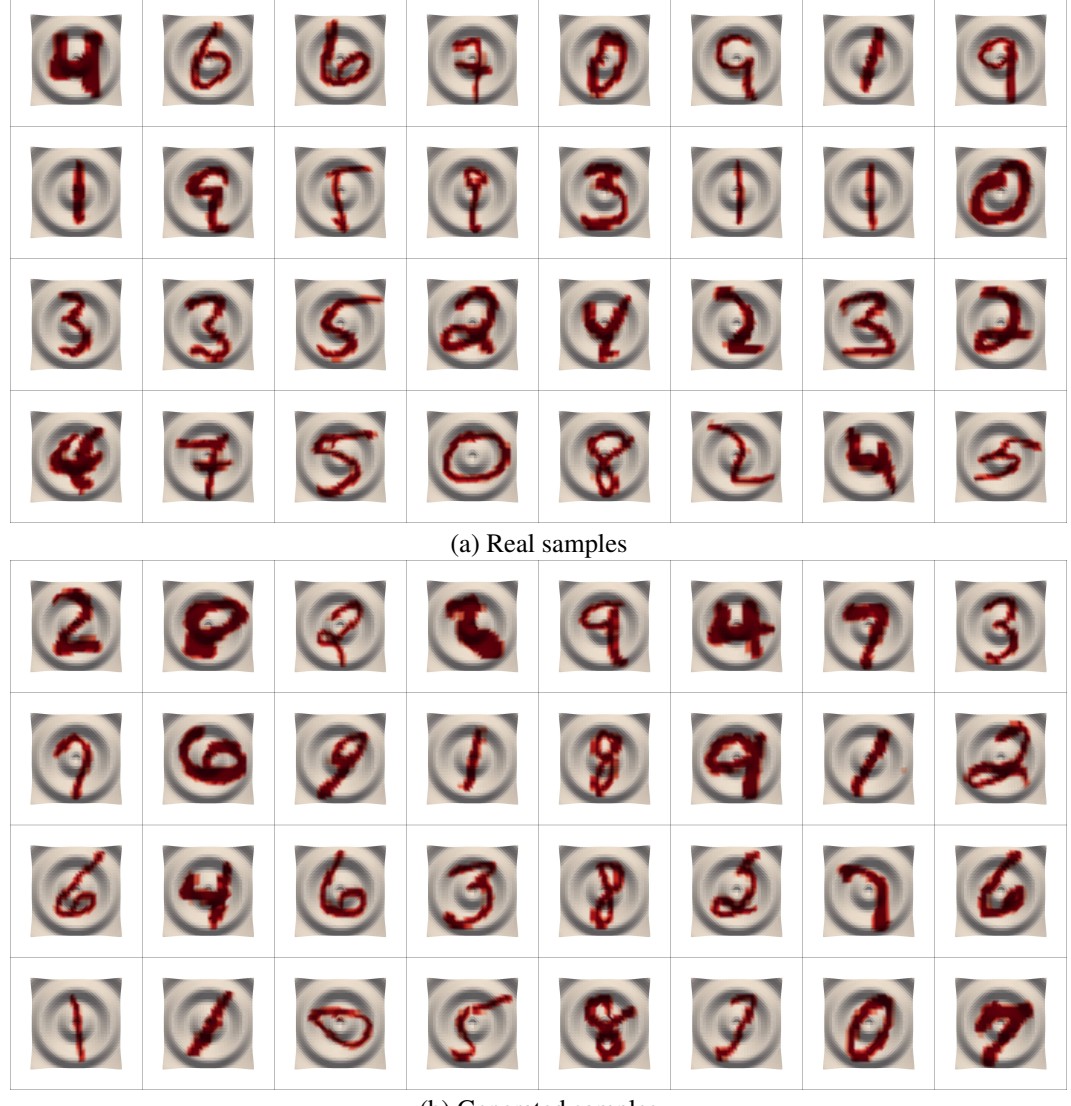

(a) Real samples

(b) Generated samples

Figure 18: Real and generated samples for MNIST (LeCun et al., 1998) digits on the wave manifold.

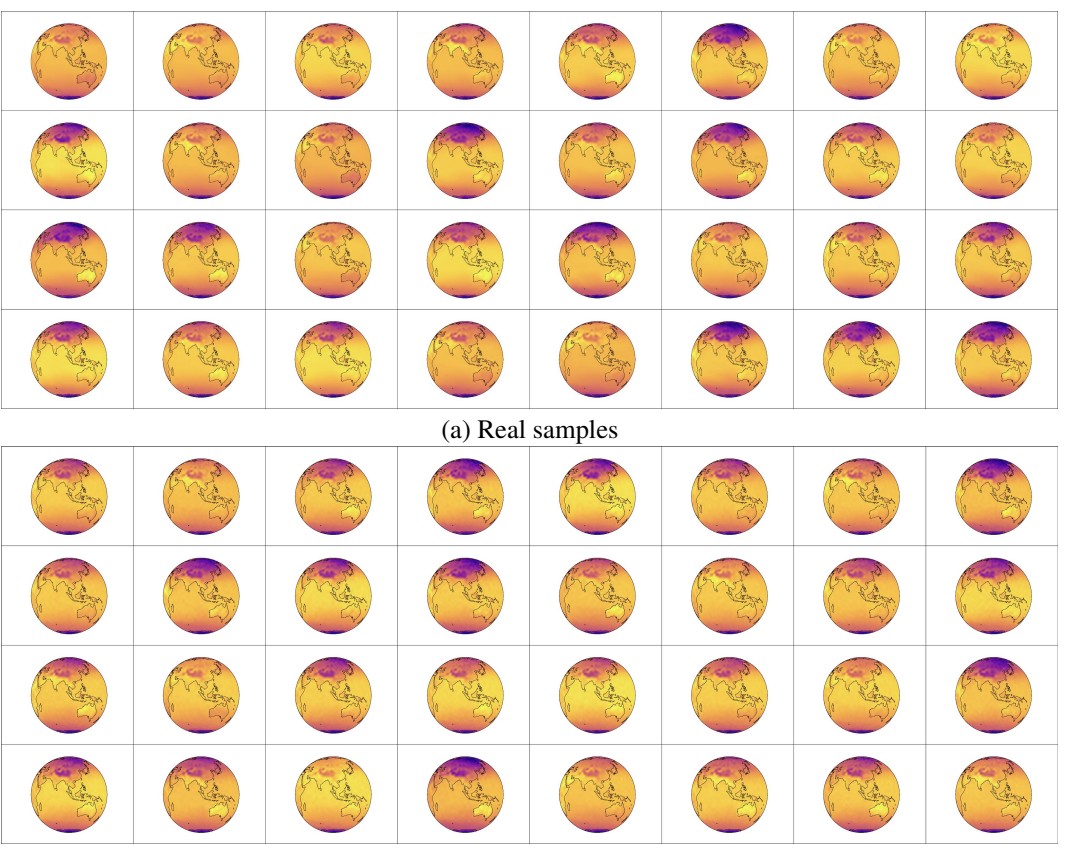

(a) Real samples

(b) Generated samples

Figure 19: Real and generated samples for the ERA5 (Hersbach et al., 2020) climate dataset.

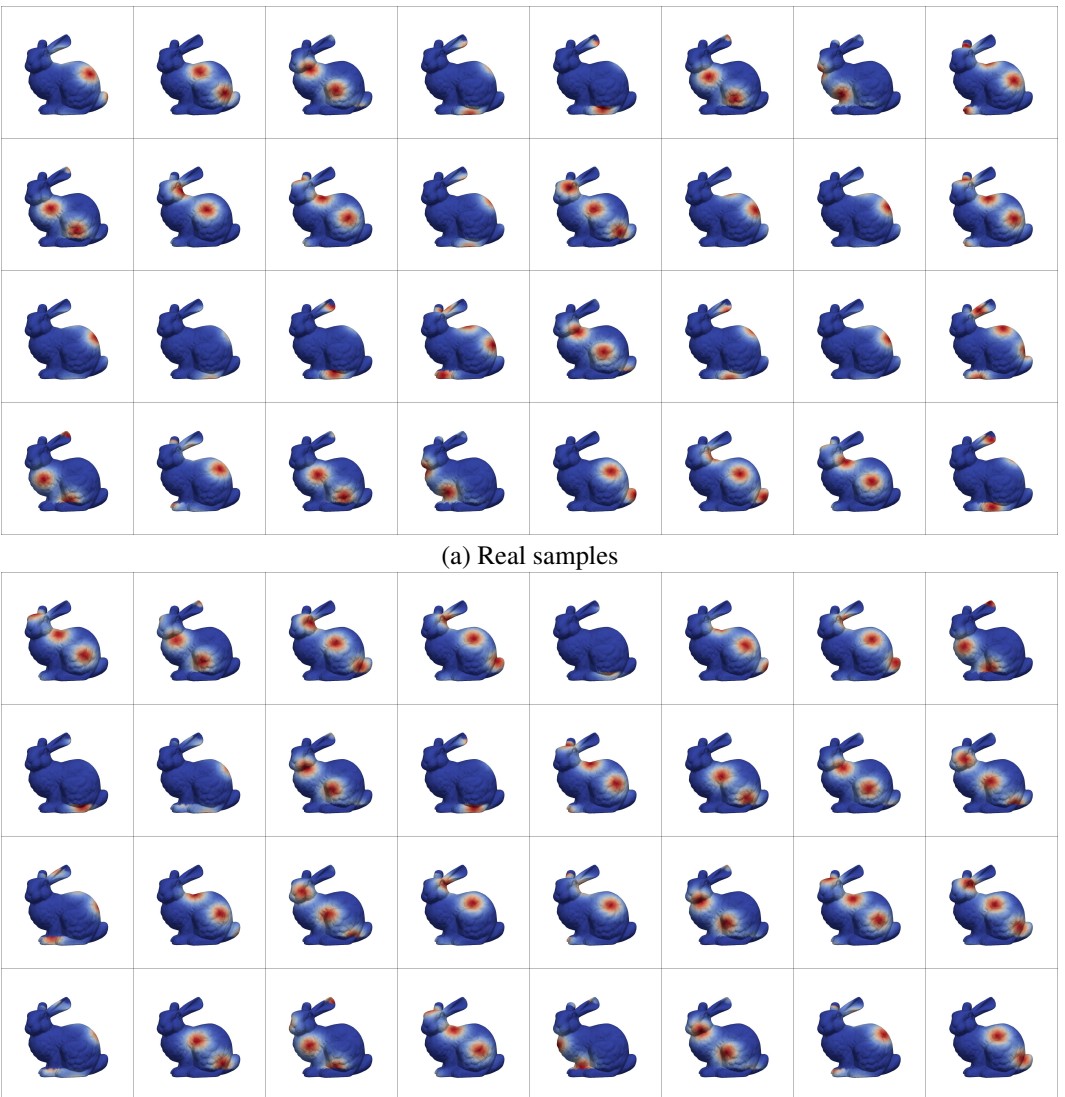

(a) Real samples

(b) Generated samples

Figure 20: Real and generated samples for the GMM dataset on the Stanford bunny manifold.

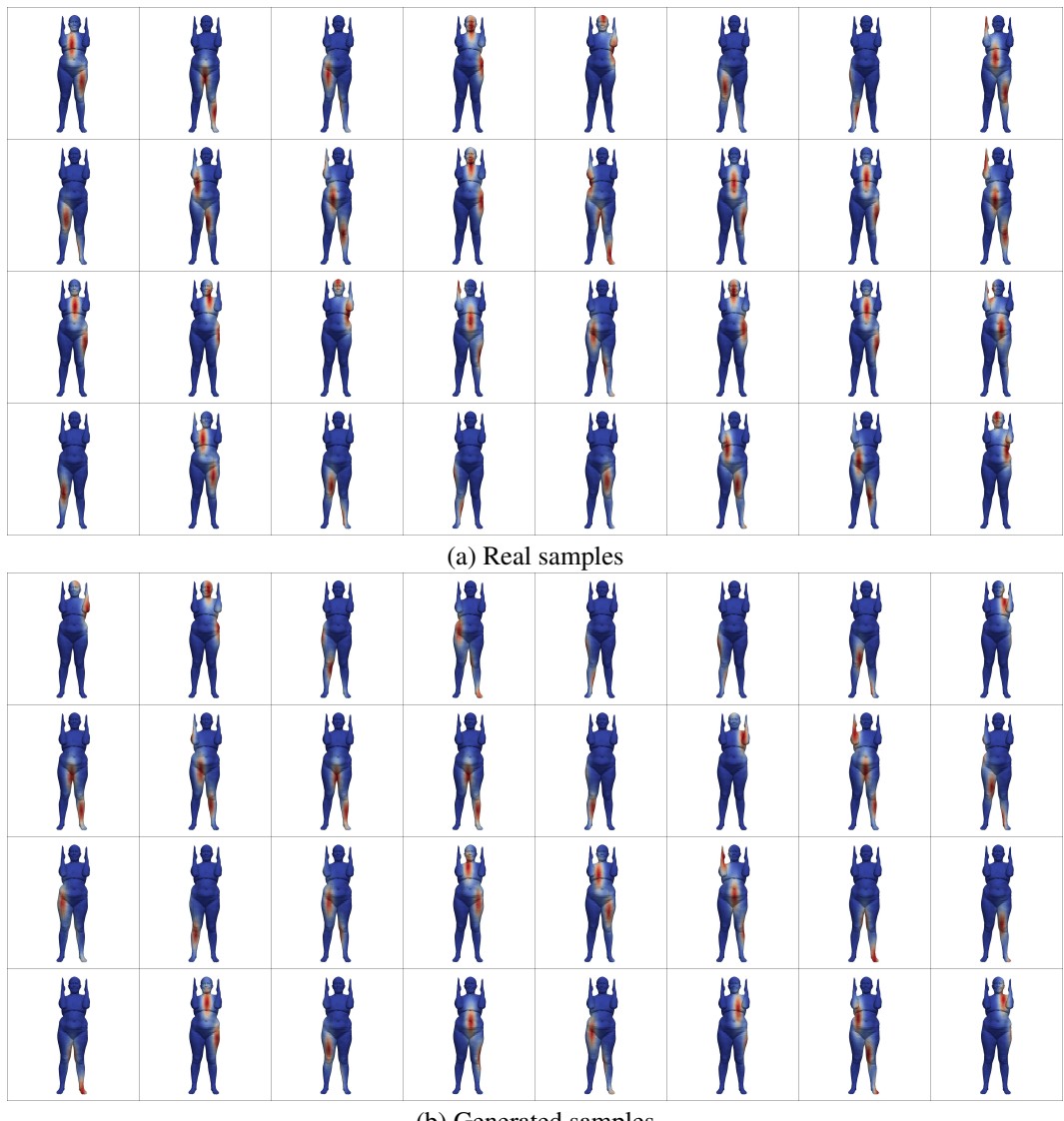

(a) Real samples

(b) Generated samples

Figure 21: Real and generated samples for the GMM dataset on the human manifold (Bronstein et al., 2008).

