# OpenReview forum: "Manifold Diffusion Fields"
_ICLR.cc/2024/Conference — ICLR 2024 poster_

### Official Review · Reviewer_wDdD · 2023-10-28

**Soundness:** 2 fair
**Presentation:** 3 good
**Contribution:** 2 fair
**Rating:** 6
**Confidence:** 4

**Summary:**

The authors extend diffusion fields onto Riemannian manifolds. They leverage the Laplace Beltrami operator as a way to get local coordinates. The authors then present experimental results in many situations comparing their approach to SOTA techniques.

Minor comment: in 3.3 end of the first paragraph should use langle rangle rather than less than greater than signs,

**Strengths:**

The overall idea the paper presents is relatively simple to understand so it is easy to follow what the authors are attempting to accomplish.

**Weaknesses:**

There are some errors when introducing Reimannian manifolds, for instance the authors state $g:\mathcal{M}\times\mathcal{M}\rightarrow\mathbb{R}_+$ but this is not true, the metric tensor takes in inputs from the tangent spaces and further can be 0.

The authors have a specific goal in mind for the manifolds they wish to consider but the theory seems to be more general in a sense. That is, typical Riemannian manifolds such as the cone of SPD matrices or a Grassman manifold go largely un mentioned. The authors focus on specific manifolds which could be interesting such as the Stanford bunny, but I lack to see the importance/applicability of it. That being said, it is a nice application but a little mysterious as to its usefulness.

**Questions:**

I understand the usefulness of using function and field interchangeably but I am curious as to why this choice was made. I ask this because over Riemannian manifolds it's common to use gradient vector fields which are defined over tangent spaces and the use of fields can cause confusion here.

---

> ### Author Response · Authors · 2023-11-15
>
> We want to thank the reviewer for their suggestions which we believe helped improve the quality of the paper. We remain open to address any further concerns that the reviewer may have to improve the paper before updating their decision.
>
> * **There are some errors when introducing Reimannian manifolds, for instance the authors state $g:\mathcal{M} \times \mathcal{M} \rightarrow \mathbb{R}_+$  but this is not true, the metric tensor takes in inputs from the tangent spaces and further can be 0.**
>
>    * We thank the reviewer for pointing this typo, we have updated the notation to include the tangent spaces. In addition, we have fixed the fact that the metric can be 0.
>
>
> * **The authors have a specific goal in mind for the manifolds they wish to consider but the theory seems to be more general in a sense. That is, typical Riemannian manifolds such as the cone of SPD matrices or a Grassman manifold go largely unmentioned.**
>
>     * We thank the reviewer for pointing this out. In this paper, we primarily provide a recipe for learning generative models over Riemannian manifolds in practical settings. As a result, our extensive experimental evaluation includes experiments in graphs, meshes and pointclouds as discrete approximations of continuous Riemannian manifolds. We are excited about the opportunities MDF affords to future work focusing on Grassman and and other analytical manifolds. This could lead to new insights on how to a model a broader class of subspaces, which could be relevant to topology but also applications in computer vision, signal processing, etc.
>
>
> * **The authors focus on specific manifolds which could be interesting such as the Stanford bunny, but I lack to see the importance/applicability of it. That being said, it is a nice application but a little mysterious as to its usefulness.**
>
>    * We thank the reviewer for pointing this out. In our submission, we focus on practical scenarios where MDF can be applied. The Stanford bunny and the other 3D meshes we used in our paper are meant to exemplify the generality of the approach on general manifolds typically used in the community. In addition, we show state-of-the-art results not only in climate prediction on the ERA5 dataset (climate on the sphere) in Sect. 5.1, but also on molecular conformer generation in Sect 5.3 (a core problem in computation chemistry). Finally, in the appendix Sect. A.6.3 we provide results using MDF to model PDEs, which is a fundamental problem across multiple scientific disciplines. We believe these experiments showcase extremely useful and exciting applications of our model across different scientific domains. In addition, MDF presents an exciting opportunity to unify diffusion generative models across many different modalities and problems.
>
>
>
> * **I understand the usefulness of using function and field interchangeably but I am curious as to why this choice was made. I ask this because over Riemannian manifolds it's common to use gradient vector fields which are defined over tangent spaces and the use of fields can cause confusion here.**
>
>    * In the area of generative models for functions, works like Functa[1], GASP[2] and DPF [3] denote functions as fields or implicit representations. We borrow this terminology for consistency since these works are the conceptually closest approaches to MDF. Note that we explicitly have avoided the use of the term "gradient vector field" or "gradient field" in the paper to not confuse readers. We have added a clarification sentence in the paper.
>
> *  **Minor comment: in 3.3 end of the first paragraph should use langle rangle rather than less than greater than signs.**
>    * We have fixed this typo.
>
>
> [1] Dupont, Emilien, et al. "From data to functa: Your data point is a function and you can treat it like one." arXiv preprint arXiv:2201.12204 (2022).
>
> [2] Dupont, Emilien, Yee Whye Teh, and Arnaud Doucet. "Generative models as distributions of functions." arXiv preprint arXiv:2102.04776 (2021).
>
> [3] Zhuang, Peiye, et al. "Diffusion probabilistic fields." The Eleventh International Conference on Learning Representations. 2022.

---

> > ### Comment · Reviewer_wDdD · 2023-11-21
> >
> > Thank you very much for your responses.
> >
> > I still question the usefulness of the particular manifolds chosen but I see the authors state they are common examples in the literature. If that is the case, that is fine. I do agree the authors have demonstrated their techniques on a variety of manifolds.

---

> > > ### Author Response · Authors · 2023-11-21
> > > **Official Comment by Authors**
> > >
> > > Thanks for engaging with us to improve the paper! We will include a discussion explaining the choice of the particular meshes used to validate MDF and limitations, and will point out that an exciting future direction is to employ MDF on a variety of problems in 3D manifolds, e.g. weather forecasting on the earth sphere [1] and PDE simulations on unstructured meshes [2], for which MDF could be directly employed.
> > >
> > > Besides, we would like to highlight that MDF is not limited to specific manifold chosen. In molecular conformation generation, a task that we model as mapping from multiple different graph manifolds to Euclidean space, MDF achieves state-of-the-art performance. It shows the potential of applying MDF to solve the essential molecular conformer generation problem in computational chemistry.
> > >
> > > Let us know if there is anything additional that can improve the paper, increase its likelihood of acceptance, and ensure it is well received.
> > >
> > > References:
> > >
> > > [1] Lam, R. et al. (2023). Learning skillful medium-range global weather forecasting. Science. doi:https://doi.org/10.1126/science.adi2336.
> > >
> > > [2] Li, Z., Huang, D.Z., Liu, B. and Anandkumar, A., 2022. Fourier neural operator with learned deformations for pdes on general geometries. arXiv preprint arXiv:2207.05209.

---

> > > > ### Author Response · Authors · 2023-11-22
> > > > **Official Comment by Authors**
> > > >
> > > > As the rebuttal deadline is approaching, we are eagerly looking forward to the your reply. We believe we have carefully addressed all the concerns and improve the quality of the paper accordingly. We hope the reviewer consider raising the score.

---

> > > > > ### Comment · Reviewer_wDdD · 2023-11-22
> > > > >
> > > > > I appreciate the follow up. Your follow up did clear up some questions I had. That being said I think my score accurately reflects the paper.

---

### Official Review · Reviewer_YGhV · 2023-10-31

**Soundness:** 3 good
**Presentation:** 3 good
**Contribution:** 2 fair
**Rating:** 6
**Confidence:** 3

**Summary:**

The paper proposed the extension of diffusion probabilistic fields onto Riemannian manifolds. The proposed method utilizes the Laplace-Beltrami operator to build an efficient representation of the points on Riemannian manifolds, which makes the proposed method invariant with respect to rigid and isometric transformations of the manifold. Numerical experiments are provided to show the efficacy of the proposed method.

**Strengths:**

The proposed method is the first work that can handle the diffusion process where the support of the distribution is on Riemannian manifolds. The proposed usage of the eigen-functions of Laplace-Beltrami operator makes the proposed method invariant w.r.t. affine and isometric transformations. The proposed method outperforms existing diffusion models on manifolds.

**Weaknesses:**

(Please respond to the questions section directly) The proposed method seems to be a plain extension of a previous work of diffusion probabilistic field. This extension shows a great potential for non-Euclidean domain but the work seems not to focusing on the discussion of how this new representation improves the work quantitatively.

**Questions:**

1. The main concern, as stated in the Weakness section, is the comparison of this work with [1]. To me, the difference of this work with [1] is only the different ways of representing points on the Riemannian manifold, where [1] directly use some coordinate since the manifolds we consider are always embedded, and this work uses the leading eigen-function of the LBO to approximate the points. It remains not very clear why and how LBO sign-functions outperforms a plain coordinate approach, or a local chart approach (since for every point on the manifold we have a local coordinate chart). Calculating eigen-functions of LBO potentially also consumes time. A more elaborative discussion of different approaches to do the embedding seems necessary.

2. The authors choose the top k LBO eigen-functions to do the embedding. is there a guidance on how to determine k?

3. Regarding Algorithm 2: First, what’s $M_q$, which seems to exist in [1] but not here? Second why do we need a random subset $C_{t}$? I thought we just need the same context and query sets as in Algorithm 1.

4. Regarding the numerical experiments, I’m particularly interested in the ERA5 dataset, which seems to be the only natural instance of manifold-supported distributions. It would be interesting to see how the proposed algorithm performs in terms of COV and MMD for this dataset, comparing to other algorithms as DPF and GASP.

References:
[1] Zhuang, Peiye, et al. "Diffusion probabilistic fields." The Eleventh International Conference on Learning Representations. 2022.

---

> ### Author Response · Authors · 2023-11-15
>
> * **To me, the difference of this work with [1] is only the different ways of representing points on the Riemannian manifold, where [1] directly use some coordinate since the manifolds we consider are always embedded, and this work uses the leading eigen-function of the LBO to approximate the points. It remains not very clear why and how LBO eigen-functions outperforms a plain coordinate approach, or a local chart approach (since for every point on the manifold we have a local coordinate chart)[...]**
>
>
>    * We thank the reviewer for raising this point, which deserves clarification. In practical terms, Tab. 1-2-3-4 in the paper show a comparison between MDF (which uses the eigen-functions of the LBO) and DPF (which uses a plain coordinate approach). Our results, show that MDF consistently outperforms DPF (ref. Tab. 1-2-3-4). We attribute this to the fact that representing points through the eigen-functions of the LBO explicitly provides the model with information about the intrinsic geometry of the manifold. Furthermore, in Sect. A.6.2  in the appendix we show that DPF completely breaks under rigid (Fig. 12) or isometric (Fig. 13) transformations of the manifold, while MDF is invariant to these transformations since it uses an intrinsic coordinate system.
>
>    * In addition, note that although one needs to compute the eigen-functions of the LBO, this computation: (i) Is very efficient, since Laplacians are sparse matrices and efficient eigen-system solvers can be used. (ii) Only needs to be performed once before training, caching eigen-functions for usage during training. In practice, the cost of computing the LBO eigen-functions is negligible compared to training the diffusion model.
>
>
>     * We have included a new section in the Appendix (Sect. A.3) discussing at length the different ways to compute embeddings.
>
>
> * **The authors choose the top k LBO eigen-functions to do the embedding. Is there a guidance on how to determine $k$?**
>
>     * Conceptually speaking, $k$ is usually dependent the specific manifold under consideration and its intrinsic geometry. Manifolds with high frequency geometry typically require larger $k$. In Figure 7, we illustrate the impact of varying $k$ on performance metrics (COV and MMD). Notably, we observe satisfactory performance even for small values of $k$. In MDF, we choose $k$ experimentally but one could also choose $k$ based on the overall variance explained of the manifold (similar to PCA).
>
>
> * **Regarding Algorithm 2: First, what’s $M_q$, which seems to exist in [1] but not here? Second why do we need a random subset ? I thought we just need the same context and query sets as in Algorithm 1.**
>
>     * In [1], $M_q$,    denotes the stacked coordinates of query pairs. In our context, we use $\varphi(\mathbf{X}_{q})$ to denote the stacked eigen-function corresponding to query pairs. Note that during inference, one can select a random subset of query pairs as context. In particular, it can be equivalent to the one used during training but it does not need to be.
>
> * **Regarding the numerical experiments, I’m particularly interested in the ERA5 dataset, which seems to be the only natural instance of manifold-supported distributions. It would be interesting to see how the proposed algorithm performs in terms of COV and MMD for this dataset, comparing to other algorithms as DPF and GASP.**
>
>    * Tab 5 provides further insights on the ERA5 dataset, demonstrating that MDF outperforms GASP in terms of both COV and MMD.
>
>
> We thank the reviewer for their suggestions which we believe helped improve the quality of the paper. We remain open to address any further concerns that the reviewer may have to improve the paper before updating their decision.

---

> > ### Author Response · Authors · 2023-11-22
> > **Official Comment by Authors**
> >
> > As the rebuttal deadline is approaching, we are eagerly looking forward to the your reply. We believe we have carefully addressed all the concerns and improve the quality of the paper accordingly.

---

> > > ### Comment · Reviewer_YGhV · 2023-11-22
> > > **Ackonwledgement of the rebuttal**
> > >
> > > I thank the authors for the rebuttal and sorry for not being able to reply due to a tight schedule on my side.
> > >
> > > I decide not to change my evaluation as I believe that this work is still not significantly from the previous work of DPF. The numerical experiment may show the efficiency of the proposed method but the contribution conceptually/theoretically seems less significant.

---

### Official Review · Reviewer_WPEM · 2023-11-02

**Soundness:** 4 excellent
**Presentation:** 4 excellent
**Contribution:** 3 good
**Rating:** 8
**Confidence:** 4

**Summary:**

The authors propose a methodology for learning fields over manifolds with denoising diffusion probabilistic models. It uses an approximation of eigen-functions of the Laplace-Beltrami Operator for the manifold at hand to create an intrinsic functional basis for the manifold. This approximation is the eigendecomposition of the graph Laplacian of the mesh approximating the manifold. This formulation theoretically allows the model to learn from data that live in manifolds, even in the absence of analytical forms for the metric of the manifold and related operators.

**Strengths:**

The paper is excellently written, with very clear explanations, illustrations, and algorithms. Experimentally, the paper covers a lot of "ground" in terms of exploring the capabilities of their model. This includes comparisons against the closest-related model in multiple scenarios and comparisons against task-specific models (molecule generation).

I believe the paper does a good job at showing how this relatively direct tweak to how DPF works is a very appropriate one. This is supported by a sufficient amount of experimental evidence, which explores most practical questions about such a change in representation.

Interestingly, the paper also shows the robustness of the model to rigid transformations on the manifold, having a much more appropriate behavior in this case than the model it is mostly based on.

**Weaknesses:**

I believe the model's biggest weakness is its novelty. Theoretically, there is only a small contribution compared to DPF (Zhuang et al., 2023).

While the authors theoretically discuss the possibility to use Laplace-Beltrami Operators as the source for a functional basis on the manifolds, this is only explored as graph Laplacians, which are a discrete approximation of manifolds. Understandably, in many scenarios it is not possible to analytically solve this problem, but it is unclear how challenging it is to adapt the ideas presented here to scenarios where it is possible (or how beneficial that is). A simple fix to this issue, in my opinion, is to claim what is being extensively evaluated experimentally: graphs/meshes as an approximation of a manifold.

**Update after rebuttal**

The authors have updated their paper to clarify their claims and point out the equivalence, at least in one experiment, of their use of graph Laplacians to the analytical LBO (Table 5 shows results of this case). After reading the comments of other reviewers and the rebuttals, I am increasing my score to 8.

**Questions:**

1. Is the model's robustness to rigid transformations on the manifold a sufficient explanation for its performance for molecule generation? Is it possible that it is also robust to other types of transformations which are useful in solving this task?

*Minor comments*
- Sec. 3.3: were $<$ and $>$ the intended symbols when writing $f$ as a linear combination of the basis?
- Sec. 3.3: "making MDF strictly operate the manifold" => "on the"?
- p. 7: "available at [URL]" I think a footnote here is less disruptive for reading

---

> ### Author Response · Authors · 2023-11-15
>
> We want to thank the reviewer for their suggestions which we believe helped improve the quality of the paper. We remain open to address any further concerns that the reviewer may have to improve the paper before updating their decision.
>
>
> * **I believe the model's biggest weakness is its novelty. Theoretically, there is only a small contribution compared to DPF (Zhuang et al., 2023).**
>
>    * In this work we generalize the problem setting in DPF to deal with functions on Riemannian manifolds. Our proposed approach is conceptually simple to implement while being very powerful in its application.  Part of our contribution was to study how far a small change in the representation of points can take the performance of the model. Using the eigen-functions of the LBO is critical in the setting. As a result the novelty aspect is in understanding what are key ingredients needed to extend in DPF.
>
>    * It is also worth noting that MDF not only outperforms DPF in the Riemannian setting, but also can be directly applied to important scientific problems ranging from climate prediction to molecular conformation obtaining state-of-the-art results. We believe that approaches that are both simple to implement and powerful tend to have long lasting impact in the field. Finally, MDF opens up an interesting area of research to unify different data modalities and problems under a common training recipe and model. For example, unifying 2D and 3D representations.
>
>
>
> * **While the authors theoretically discuss the possibility to use Laplace-Beltrami Operators as the source for a functional basis on the manifolds, this is only explored as graph Laplacians, which are a discrete approximation of manifolds. Understandably, in many scenarios it is not possible to analytically solve this problem, but it is unclear how challenging it is to adapt the ideas presented here to scenarios where it is possible (or how beneficial that is). A simple fix to this issue, in my opinion, is to claim what is being extensively evaluated experimentally: graphs/meshes as an approximation of a manifold.**
>
>
>    * We thank the reviewer for raising this point which we completely agree deserves clarification. We have clarified in the contributions that the bulk of our experimental evaluation covers graphs, meshes and pointclouds (ref. Tab 6 and Fig. 8) as discrete approximations of manifolds, which is important in most practical settings.
>
>    * We would also like to point the reviewer to Tab 5. where we show results of MDF on the ERA5 dataset, which consists of temperatures on on a parametric manifold (eg. earth represented as a sphere). In this case, the analytical solution for the eigenfunctions of the LBO are given by spherical harmonics (second paragraph of page 7). Our results show that using spherical harmonics MDF outperforms previous approaches to learn functions on the sphere, like GASP [2]. We have clarified this in the paper.
>
>    * A final clarification point is that this approach has a theoretical connection to the analytical LBO, as the eigenvectors and eigenvalues of the graph Laplacian converge to the eigenfunctions and eigenvalues of the analytical LBO as the number of vertices/nodes tends to infinity [1].
>
>     [1] Convergence of Laplacian eigenmaps. in Adv. Neur. In.: Proceedings of the 2006 Conference, vol. 19, p. 129. The MIT Press.
>
>     [2] Dupont, Emilien, Yee Whye Teh, and Arnaud Doucet. "Generative models as distributions of functions." arXiv preprint arXiv:2102.04776 (2021).
>
> * **Is the model's robustness to rigid transformations on the manifold a sufficient explanation for its performance for molecule generation? Is it possible that it is also robust to other types of transformations which are useful in solving this task?**
>
>   * We thank the reviewer for raising this very interesting point. It is true that the robustness to rigid transformations can be helpful for conformer generation, we do believe that this is not the only explanation for the performance of MDF. Our current hypothesis is that the eigen-functions of the LBO provide a function basis that helps the model find representations that can be aligned across molecules and in turn this boosts performance. Note that generating conformers can be thought of in a way as a non-rigid transformation of a base molecular structure and MDF appears to be quite good at discovering this set of non-linear symmetries. We will fully uncover and understanding of this behaviour in future work.
>
>
>
> * **Sec. 3.3: were < and > the intended symbols when writing  as a linear combination of the basis?**
>    * Thanks for pointing out this typo, we have fixed it in the new version of the paper.
>
> * **Sec. 3.3: "making MDF strictly operate the manifold" => "on the"?**
>    * We fixed the typo in the new version of the paper.
>
> * **p. 7: "available at [URL]" I think a footnote here is less disruptive for reading**
>    * We have moved this into the appendix to keep the paper at 9 pages.

---

> > ### Comment · Reviewer_WPEM · 2023-11-22
> >
> > > We believe that approaches that are both simple to implement and powerful tend to have long lasting impact in the field.
> >
> > I 100% agree with that. The point I was making is rather on the novelty of the theoretical contribution. This would be the biggest weakness not because it is a bad thing, but because in other aspects the paper is strong. I believe, experimentally, the paper shows more than enough evidence that this theoretically simple change is indeed impactful and I appreciate that.
> >
> > > We have clarified in the contributions that the bulk of our experimental evaluation covers graphs, meshes and pointclouds (ref. Tab 6 and Fig. 8) as discrete approximations of manifolds, which is important in most practical settings.
> >
> > Alright! I'm happy with that.
> >
> > > We would also like to point the reviewer to Tab 5. where we show results of MDF on the ERA5 dataset, which consists of temperatures on on a parametric manifold (eg. earth represented as a sphere). In this case, the analytical solution for the eigenfunctions of the LBO are given by spherical harmonics (second paragraph of page 7). Our results show that using spherical harmonics MDF outperforms previous approaches to learn functions on the sphere, like GASP [2].
> >
> > Thank you for pointing out this detail I overlooked, the clarification, and updating the paper to make this evident. I am more than pleased to update my score.
> >
> > > It is true that the robustness to rigid transformations can be helpful for conformer generation, we do believe that this is not the only explanation for the performance of MDF. Our current hypothesis is that the eigen-functions of the LBO provide a function basis that helps the model find representations that can be aligned across molecules and in turn this boosts performance. Note that generating conformers can be thought of in a way as a non-rigid transformation of a base molecular structure and MDF appears to be quite good at discovering this set of non-linear symmetries.
> >
> > That seems reasonable to me. I am curious to see how exploring this turns out.

---

> > > ### Author Response · Authors · 2023-11-22
> > > **Official Comment by Authors**
> > >
> > > We thank the reviewer for raising the score and championing our work. We appreciate the reviewer for the valuable comments that help substantially improve the quality of the manuscript.

---

### Author Response · Authors · 2023-11-15
**To all reviewers and ACs**

We thank all the reviewers for their time and valuable feedback, which helped in highlighting our contributions and improving our work. We have updated the manuscript, highlighting the changes in red color.  Here's a summary of points highlighted by the reviewers:


- WPEM:  The authors propose a methodology for learning fields over manifolds with denoising diffusion probabilistic models. [...] This formulation theoretically allows the model to learn from data that live in manifolds, even in the absence of analytical forms for the metric of the manifold and related operators. Experimentally, the paper covers a lot of "ground" in terms of exploring the capabilities of their model. This includes comparisons against the closest-related model in multiple scenarios and comparisons against task-specific models (molecule generation).

- YGhV: The paper proposed the extension of diffusion probabilistic fields onto Riemannian manifolds. The proposed method utilizes the Laplace-Beltrami operator to build an efficient representation of the points on Riemannian manifolds, which makes the proposed method invariant with respect to rigid and isometric transformations of the manifold. The proposed method is the **first work** that can handle the diffusion process where the support of the distribution is on Riemannian manifolds.

- wDdD: The authors extend diffusion fields onto Riemannian manifolds. They leverage the Laplace Beltrami operator as a way to get local coordinates. The authors then present experimental results in many situations comparing their approach to SOTA techniques.



We note that the reviewers describe our work as a new methodology for learning generative models of fields over manifolds with diffusion generative models and also is the **first work** that can handle the diffusion process of functions defined on Riemannian manifolds. We show several empirical results over multiple challenging problems like climate prediction on the sphere or molecular conformations over graphs. Finally, MDF opens a new path for applying the diffusion generative models to dynamical systems, like PDEs on manifolds in Sect. A.6.3.

---

### Author Response · Authors · 2023-11-21

Dear reviewers,

Thanks for your time and commitment reviewing our submission, we believe your comments have helped improve the quality of the paper! We have addressed reviewers comments and highlighted changes in the manuscript in red color.

Since we are into the last two days of the discussion phase, we are eagerly looking forward to your post-rebuttal responses. Please do let us know if there are any additional suggestions to improve the paper such that it can be accepted.

Authors

---

### Meta-Review · Area_Chair_obch · 2023-12-08

**Metareview:**

The paper propose a method for learning fields over manifolds with denoising diffusion probabilistic models. It uses an approximation of eigenfunctions of the Laplace-Beltrami Operator on the manifold to define a coordinate system on the manifold. The theoretical formulation is accompanied by numerical experiments.

Reviewers point out that the proposed method has limited theoretical novelty compared with the previous work "Diffusion probabilistic fields" (DPF) of  Zhuang et al. (ICLR 2023). The main difference is  the coordinate system being used. However, they also appreciate that experimentally, the current method outperforms previous methods such as DPF.

Note:
- The sentence "Riemannian manifolds are connected and compact manifolds" is not correct. You are assuming a connected, compact Riemannian manifold here.

**Justification For Why Not Higher Score:**

The theoretical contribution is limited compared to the previous work "Diffusion probabilistic fields" (DPF) of  Zhuang et al. (ICLR 2023)

**Justification For Why Not Lower Score:**

Experimentally, the proposed method shows improvements over existing methods, including the DPF

---

### Decision · Program_Chairs · 2024-01-16

Accept (poster)